

# Organic matters, but inorganic matters too: column examination of elevated mercury sorption on low organic matter aquifer material using concentrations and stable isotope ratios.

David S. McLagan[1,2,3,*], Carina Esser[1,*], Lorenz Schwab[4,5], Jan G. Wiederhold[4], Jan-Helge Richard[6], Harald Biester[1].

- Institute of Geoecology, Technische Universität Braunschweig, Braunschweig, 38106, Germany.
- Department of Geological Sciences and Geological Engineering, Queen's University, Kingston, ON, K7L3N6, Canada.
- School of Environmental Studies, Queen's University, Kingston, ON, K7L3J6, Canada.
- Department for Environmental Geosciences, Centre for Microbiology and Environmental Systems Science, University of Vienna, Vienna, 1090, Austria.
- Environmental Engineering Institute IIE-ENAC, Soil Biogeochemistry Laboratory, Ecole Polytechnique Fédérale de Lausanne (EPFL), Sion, Switzerland.
- Institute for Hygiene and Environment Hamburg, 20539 Hamburg, Germany

* - These authors contributed equally to the manuscript.

correspondence to: David McLagan, david.mclagan@queensu.ca    and

Harald Biester, h.biester@tu-braunschweig.de





## Abstract

Sorption of mercury (Hg) in soils is suggested to be predominantly associated with organic matter
(OM). However, there is a growing collection of research that suggests clay minerals and Fe/Mn-
oxides are also important solid-phases for the sorption of soluble Hg in soil-groundwater systems.
We use a series of (60 mL syringe based) column experiments to examine sorption and subsequent
desorption of $HgCl_2$ solutions (Experiment 1 [EXP1]: 46.1 ± 1.1 mg $L^{-1}$; and Experiment 2 [EXP2]: 144
± 6 mg $L^{-1}$) in low OM (0.16 ± 0.02 %) solid-phase aquifer materials. Analyses of total Hg
concentrations, Hg speciation (i.e., pyrolytic thermal desorption (PTD)), and Hg stable isotopes are
performed on both solid- and liquid-phase samples across sorption and desorption phases. Sorption
breakthrough curve best fitted a Freundlich model. Despite the very low OM content, the Hg
equilibrium sorptive capacity in these columns is very high: 1510 ± 100 and 2320 ± 60 mg $kg^{-1}$ for
the EXP1 and EXP2, respectively, and is similar to those determined for high OM soils. Desorption
fits exponential decay models and 46 ± 6% and 58 ± 10% of the sorbed Hg is removed from the solid-
phase materials at the termination of desorption in EXP1 and EXP2, respectively. This desorption
profile is linked to the initial release of easily exchangeable Hg(II) species physically sorbed to
Fe/Mn-oxides and clay mineral surfaces and then slower release of Hg(II) species that have
undergone secondary reaction to more stable/less soluble Hg(II) species and/or diffusion/transport
into the mineral matrices. Hg stable isotope data support preferential sorption of lighter isotopes
to the solid-phase materials with results indicating isotopically heavy liquid-phase and isotopically
light solid-phase. The divergence of $\delta^{202}$Hg (describing mass dependent fractionation (MDF))
between liquid- and solid-phase continues into desorption and we attribute this to lighter isotopes
being favoured in secondary processes occurring after initial sorption to the solid-phase materials
(i.e., matrix diffusion, change in Hg(II) speciation, elemental Hg (Hg(0)) production) that lead to less
exchangeable forms of Hg. Consequently, heavy isotopes are preferentially released during
desorption. These observations agree with data from $HgCl_2$ contaminated sites. The secondary
production of Hg(0) within the columns is confirmed by PTD analyses that indicate distinct Hg(0)
release peaks in solid-phase samples at <175 °C, which again agree with field observations.
Retardation ($R_D$) and distribution ($K_D$) coefficients are 77.9 ± 5.5 and 26.1 ± 3.0 mL $g^{-1}$ in EXP1,
respectively, and 38.4 ± 2.7 and 12.4 ± 0.6 mL $g^{-1}$ in EXP2, respectively. These values are similar to
values derived from column experiments on high OM soil and provide the basis for future Hg fate
and transport modelling in soil-groundwater systems.
**Keywords:** Mercury stable isotopes, column experiments, sorption/desorption, groundwater,
polluted sites, distribution coefficient.

## 1   Introduction

Mercury (Hg), a transition metal of group 12 and period 6 of the periodic table, has a unique
electrochemical structure. The pair of electrons in the outermost (6s) shell have a relativistically
contracted radius, which greatly reduces the element's ability to form metal-metal bonds (Norrby,
1991). Hence, Hg is the only liquid-phase metal at standard temperature and pressure. Even with
this radial contraction, Hg is an atomically large element, and species in its divalent oxidation state
qualify as "soft-acids", which under hard and soft Lewis acid and base theory results in Hg having
greater affinity for "soft-bases" (Ho, 1975). One particularly pertinent "soft-base" for Hg is sulphur.
Cinnabar (α-HgS) and meta-cinnabar (β-HgS) are the dominant forms of Hg in the lithosphere
(Gettens et al., 1972; Clarkson, 1997), but are relatively stable ores, have very low solubility, and





low bioavailability (Llanos et al., 2011; Lu et al., 2011). Mining of these cinnabar ores for industrial
use of Hg has heavily perturbed the natural biogeochemical cycle of Hg. Other primary sources of
Hg emissions/releases to the environment include geogenic (natural), fossil-fuel combustion,
industrial and medical uses of Hg, and legacy emissions from Hg polluted sites (Pirrone et al., 2010;
Kocman et al., 2013; Streets et al., 2019).
While redox conditions and organic matter (OM) availability and composition are key determinants
in the mobility of Hg in aquatic/saturated subsurface environments, pH (Andersson, 1979; Gu et al.,
2011; Manceau and Nagy, 2019), chloride concentration ($Cl^-$; Schuster, 1991), and speciation of Hg
inputs (particularly for polluted systems; McLagan et al., 2022) also play important roles. Solubilities
of Hg species vary widely from practically insoluble cinnabar species ($\approx 2*10^{-24}$ g $L^{-1}$) to low solubility
elemental Hg (Hg(0): $\approx 5*10^{-5}$ g $L^{-1}$) to highly soluble Hg(II)-chloride ($HgCl_2$) (66 g $L^{-1}$) (Sanemasa,
1975; Schroeder and Munthe, 1998; Skyllberg et al., 2012). In systems that are OM limited, clay
minerals and oxides, hydroxides, and oxyhydroxides of Fe, Mn and Al become increasingly important
sorbents for Hg species (Lockwood and Chen, 1973; Schuster 1991; Kim et al., 2004). Additionally,
there is a strong tendency of Hg(II) to complex with hydroxides and halides under oxic conditions
(Schuster, 1991, Ullrich et al., 2001). Uptake of Hg to inorganic sorbents has been reported to occur
via rapid initial surface sorption followed by slower phase of Hg undergoing secondary
transformation to more stable/less soluble species or diffusing into the mineral matrices (Avotins,
1975; Miretzky et al., 2005; McLagan et al., 2022).
More recently, laboratory and field studies have expanded biogeochemical assays of Hg in
subsurface environments using stable isotopes (Jiskra et al., 2012; Zheng et al., 2018; McLagan et
al., 2022). Hg is an isotopic system that has seven stable isotopes and to which environmental
processes can impart mass-dependent (MDF) as well as both odd and even mass-independent (MIF)
fractionation (Bergquist and Blum, 2007; 2009; Wiederhold, 2015). In particular, this capacity for Hg
stable isotope analyses to elicit valuable information on tracing/identifying specific environmental
processes make them a vital tool in the examination of Hg biogeochemical cycling (Bergquist and
Blum, 2007; 2009; Wiederhold, 2015).
Traditionally, column and batch experiments have been utilised to assess the sorption (including
sorption or distribution coefficient: $K_D$ and the related retardation coefficient: $R_D$) and mobility of
contaminants for solid-phase soil and aquifer materials. Both methods have strengths and
weaknesses. Batch experiments represent the simplest means to test analyte sorption, but these
experiments are static, and equilibrium oriented; questions about the applicability of the results to
natural systems with flowing water and potentially changing levels of saturation logically persist
(Schlüter et al., 1995 Schlüter, 1997; Van Glubt et al., 2022). Flow-through columns provide a much
more dynamic and manipulatable experimental environment that is also not exclusively limited to
equilibrium-based sorption simulations. Nonetheless, they are more laborious, difficult to replicate
from column to column, column boundaries (walls) can present preferential flow problems, and
despite the ability to manipulate the physicochemical properties of the columns this inevitably
underrepresents the inherent variability of actual soil/aquifer conditions (Sentenac et al., 2001;
USEPA, 2004). Soil contaminant transport modelling is a rapidly developing field of research and
provides an alternative/complementary method to these traditional experimental methods. While
Hg soil transport modelling is also advancing, progress is somewhat limited by the lack of
measurement data particularly relating to $K_D$ values, Hg speciation and methods of assessing specific
processes for different soil/solid-phase materials (Leterme et al., 2014; Richard et al., 2016a).




Thus, it is important from both experimental and modelling standpoints that we determine effective
means of deriving information on sorption/mobility of Hg in soils. Lacking the capacity to measure
aquifer systems *in-situ*, we deem column experiments using solid-phase materials sourced from
sites of interest as the best available method to do so. Within this study, we aim to determine the
sorptive (and desorptive) capacity of low OM aquifer materials for Hg(II) using column experiments
and total Hg concentration, speciation, and stable isotope analyses of both solid and liquid-phase
materials. These experiments will be the first conducted on such low OM soil/aquifer material and
provide critical data into Hg transport and sorption within low OM soil and aquifer systems to
improve our geochemical understanding of subsurface Hg behaviour and for soil chemistry and
transport modelling. In addition, these column experiments on uncontaminated aquifer material
sourced from adjacent to a former industrial site at which $HgCl_2$ was applied as wood preservative
will simulate the contamination process. Data will aid our interpretation of the Hg biogeochemistry
in coupled soil-groundwater systems, as well as future Hg groundwater transport modelling, and
potentially provide guidance on contaminated site remediation.

## 120  2   Methods

### 121  2.1   Materials and experimental setup

The solid-phase material used in these experiments is highly permeable sand-gravel sediments
sourced from the saturated zone of an unconsolidated aquifer (approximate depth: 10 m) extracted
by a soil drill core in 2019. This site was impacted by losses of approximately 10-20 tonnes of Hg in
the form of high concentration $HgCl_2$ solution (≈0.66% $HgCl_2$) that was applied to timber as a
preservative (Schöndorf et al., 1999; Bollen et al., 2008; McLagan et al., 2022). The solid-phase
materials were extracted from outside of the plume of contaminated groundwater (Site B in
McLagan et al., 2022); and hence, the starting Hg concentration within was very low (Table 1). The
material was stored in a dark and cool place before drying at 30 °C for 48 hours. It was then sieved
to a size of <2 mm using a mesh soil sieve, which resulted in a distribution of 74.1 ± 4.6% coarse load
(>2 mm; not used) and 25.8 ± 4.6% fine load (<2 mm). A subsequent particle size analysis of the fine
load was carried out using sieving and sedimentation method (DIN ISO 11277, 2002), and results
(see Table 1) categorise the solid-phase aquifer materials as a sandy-loam on the soil texture
triangle. A summary of the properties of the investigated material is shown in Table 1.
*Table 1: Properties of the solid-phase aquifer material used.*

| Parameter | Fe (g kg$^{-1}$) | Mn (mg kg$^{-1}$) | Hg (µg kg$^{-1}$) | TC (%) | TOC (%) | TIC (%) | Clay (%) | Silt (%) | Sand (%) |
|---|---|---|---|---|---|---|---|---|---|
| **Value** | 19.2 ± 1.5 | 690 ± 160 | 20.4 ± 1.0 | 0.50 ± 0.03 | 0.16 ± 0.02 | 0.34 ± 0.03 | 13.5 | 23.2 | 63.3 |
| **Samples (*n*)** | 16 | 16 | 6 | 3 | 3 | 3 | 1 | 1 | 1 |


A set of preliminary experiments prior to experiment 1 (EXP1) and experiment 2 (EXP2) were run to
optimise packing methods, flow rates, stock solution concentration, and time the experiments
would take, and these are detailed in Section S1. Based on these preliminary data the experimental
setup was based on a modified version of DIN method 19528-01 (DIN 2009). 8x 60 mL disposable
polypropylene syringes (height: 15.49 cm; inner diameter: 2.97 cm) were used as columns in each
experiment (Figure 1). The insides of the columns were roughened with sandpaper (and thoroughly
cleaned with surfactant and rinsed with DI water to remove any debris) in order to minimise
preferential flow along the walls of the column. Each column was then filled with a layer of quartz



wool and a layer of quartz beads whose combined volume reached the 10 ml mark on the syringe.
The sieved and dried material was then transferred by ≈14 g aliquots into the syringes (preliminary
testing revealed dry packing achieved optimal column density and was best at preventing
separation). Each aliquot was compacted to the desired volume and the surface of each aliquot was
broken up before the addition of the subsequent aliquot to prevent layering between each addition.
The mean mass and bulk density ($\rho_b$) of the solid-phase aquifer materials added to the columns was
$70.09 \pm 0.04$ g and $1.42 \pm 0.01$ g cm$^{-3}$, respectively, in EXP1, and $70.05 \pm 0.03$ g and $1.43 \pm 0.01$ g cm$^{-3}$
, respectively, in EXP2. This resulted in the height of the solid-phase materials within the column
being ≈11 cm. Additional layers of quartz beads then quartz wool (syringe volume again ≈10 mL)
were added on top of the solid-phase materials to reduce column separation and particle transport.
Individual columns are names C1.1 to C1.8 for EXP1 and column C2.1 to C2.8 in EXP2.
All column experiments were conducted under saturated conditions. Figure 1A shows the
configuration of the setup with the peristaltic pump upstream of the columns and flow through the
columns was bottom to top to minimise entrapment of air and preferential flow paths. The stock
solution, peristaltic pump, columns, and eluate sampling points were connected with 3.125 mm
(inner-diameter) polypropylene tubing (length: $105 \pm 10$ cm; $n = 16$). To simulate the aquifer (flow
velocity of ≈3 – 10 m day; Schöndorf et al., 1999; Bollen et al., 2008) and prevent separation of the
solid-phase materials within the column, the lowest possible volume flow of $0.62 \pm 0.02$ ml min$^{-1}$ ($n$
= 16) was set across all columns (flow velocities measured before and after experiments; Section
S2). The stock solution was made using mixing $HgCl_2$ salt with tap water and stored in a 20 L
polyethylene container. Tap water was selected due to its inherent concentration of ions, low
potential for biological activity, and ease-of-use (challenges in extraction, storage, and transport of
large groundwater volumes from study site ≈600 km away). Critically, the tap water and eluate DOC
concentrations ($2.3 – 3.3$ mg L$^{-1}$) were of a similar range (even slightly less) than the values measured
by Richard et al (2016a) at the site these solid-phase materials were removed ($3.8 – 6.3$ mg L$^{-1}$). This
should eliminate the possibility that tap water would introduce a significant amount of artificial
sorption sites associated with DOC being added to the system. Stock solutions were $46.1 \pm 0.1$ mg
L$^{-1}$ in EXP1 ($n = 6$) and $144 \pm 6$ mg L$^{-1}$ in EXP2 ($n = 12$) and were selected as estimates of the original
concentrations of $HgCl_2$ contaminated solution entering the soil-groundwater system considering
groundwater concentrations up to $164 \pm 75.4$ µg L$^{-1}$ are still observed 55 years after closure of the
industrial activities at the site the solid-phase materials were extracted (McLagan et al., 2022). The
physicochemical properties of both the stock solutions and eluate were monitored across the
experiments and data are listed in Section S2. Desorption was performed by replacing the stock
solution with tap water flowing at the same velocity. In total (sorption, equilibrium, and desorption),
EXP1 and EXP2 ran continuously for 14 days, 3 hours, and 9 minutes, and 10 days, 13 hours and 4
minutes, respectively.
Columns were pre-conditioned with tap water for 1 week at the experimental flow velocity to allow
equilibration between the solid-phase materials and the dissolved substances in the tap water, the
major component of the stock solution used within the experiment. After 24 hours of pre-
conditioning, NaCl salt solution tracer experiments were conducted to monitor the rate of water
transfer through the columns (assuming NaCl is a conserved tracer that does not interact with the
solid-phase materials). The NaCl solution was passed into the system for 10 minutes and then
replaced with tap water. The change in conductivity was measured over time using a hand-held
electronic conductivity meter to produce NaCl (tracer) breakthrough curves. Results show good



column flow consistencies similar to the volumetric flow measurements and both data sets are
described in detail in Sections S1 and S2. The system was rigorously tested and checked for leaks
during both the pre-conditioning and testing phases.

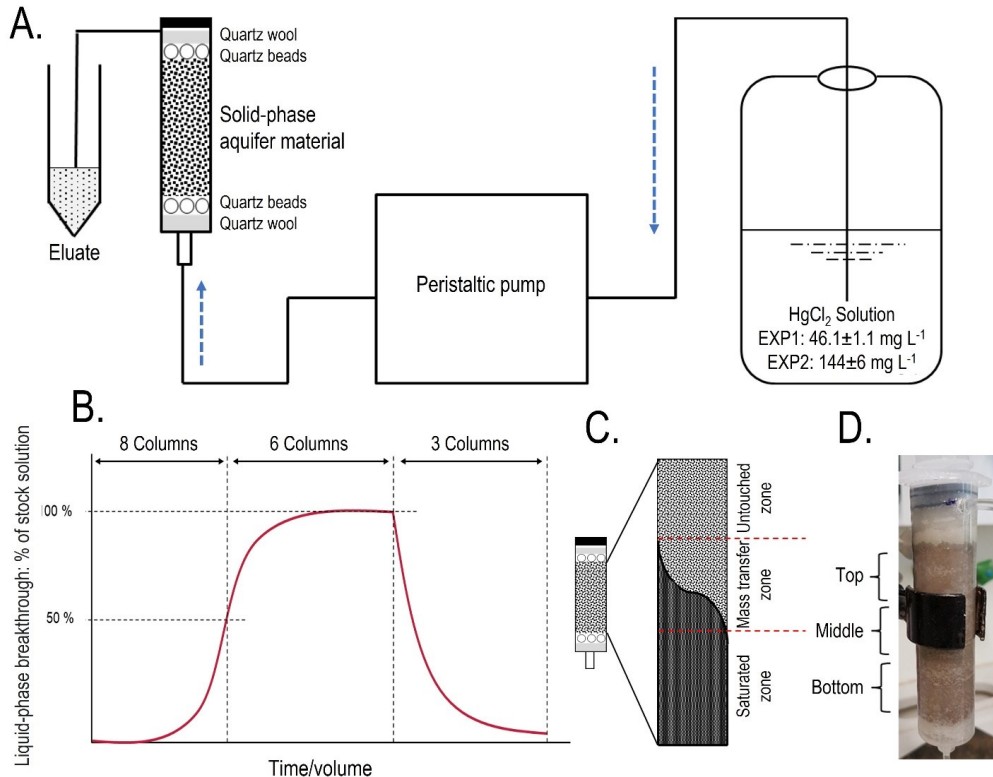


*Figure 1: A. Schematic representation of the experimental setup. B. Theoretical model of the*
*experiments indicating sorption and desorption phases and column termination points for solid*
*phase analyses (2 columns terminated at 50 % breakthrough, 3 columns terminated at ≈equilibrium,*
*and the final 3 columns terminated after desorption; end of experiment). C. Representation of the*
*zones of mass transfer of Hg during the sorption phase ("saturated zone" refers to solid-phase in*
*that zone reaching its equilibrium uptake capacity for Hg at the experimental solution*
*concentration). The dark area describes the rising front of mercury. D. Allocation of column sections*
*(≈15 mL in each section) for solid-phase analyses ("Bottom" is the solution entry point).*
10 mL of eluate was allowed to flow off into a waste vessel before sample collection periods. The
liquid-phase was sampled for total Hg (THg) concentrations consistently throughout the
experiments: 38x in EXP1 (10x up to ≈50% breakthrough – columns C1.1-C1.8; 11x between ≈50%
breakthrough and ≈equilibrium – columns C1.1-C1.6; and 17x during desorption – columns C1.1-
C1.3) and 35x in EXP2 (8x up to ≈50% breakthrough – columns C2.1-C2.8; 16x between ≈50% and
≈100% breakthrough – columns C2.1-C2.3 and C2.6-C2.8; and 11x during desorption – columns C2.1-
C2.3). Liquid-phase speciation samples were collected 8x at ≈25%, 50%, 75% breakthrough, and
≈equilibrium, at the end of the equilibrium (immediately before stock solution was changed to tap
water), and ≈0% (immediately after stock solution was changed to tap water), 50% and at the end
of desorption for both experiments. Liquid-phase stable isotope samples were collected only from



columns C2.1-C2.3 in EXP2 9x in total. Collections were similar to liquid-phase speciation sampling
points with an additional collection during the sorption stage of the experiment. After termination,
solid-phase materials were analysed for THg concentrations, Hg species, and Hg stable isotopes. In
summary, C1.7 and C1.8 and C2.4 and C2.5 were sacrificed at 50% breakthrough; C1.4-C1.6 and
C2.6-C2.8 after equilibrium (100% breakthrough); while C1.1-C1.3 and C2.1-C2.3 went through to
the end of desorption.

## 2.2    Analyses

### 2.2.1    Liquid-phase THg and speciation analyses

Eluate samples for THg and Hg stable isotope analyses were immediately stabilized by adding 1% by
volume of 0.2 M bromine monochloride (BrCl) prepared according to Bloom et al. (2003). In order
to break up all of the organically bound mercury in the liquid, a reaction time of the BrCl of 24-hours
is recommended (US EPA method 1631, 2002). However, with little OM (Table 1), we assessed
sample THg analysis only 1-hour after BrCl addition and there was no impact on sample recovery
(Table S1.2). Immediately prior to analysis, hydroxylamine hydrochloride ($NH_2OH \cdot HCl$) was added to
neutralize the BrCl followed by addition of tin(II) chloride ($SnCl_2$) solution as the Hg reducing agent.
Liquid-phase speciation analyses followed the same methods described elsewhere (Bollen et al,
2008; Richard et al., 2016b; McLagan et al., 2022). This method is described as a complementary
qualitative analytical tool and produces four distinct "fractions" of the total pool of liquid-phase Hg:
(i) elemental Hg (Hg(0)) (purged from untreated eluate sample), (ii) dissolved inorganic Hg(II) termed
Hg(II)A; (purged after reduction with $SnCl_2$ treatment; e.g. $HgCl_2$); (iii) DOM-bound Hg(II) termed
Hg(II)B (purged after BrCl and $SnCl_2$ treatment), and (iv) particulate Hg termed Hg(II)P (difference
between THg concentrations in filtered and total unfiltered eluate samples). Both concentration and
speciation results were measured using a cold-vapor atomic absorbance spectrometer (CV-AAS)
(Hg-254 NE, Seefelder Messtechnik GmbH, Germany) according to DIN method 1483 (2007) and
USEPA method 1631 (2002).

### 2.2.2    Solid-phase THg and speciation analyses

After individual columns were sacrificed for solid-phase analyses, the ends of the columns were
sealed to prevent the columns from draining and stored in the same upright position as the
experimental setup (Figure 1) to prevent further disturbance. Columns were cut into sections (Figure
1D), homogenised and subset within 1 week of the end of the experiments and stored at 4°C in
brown (opaque) falcon tubes until digestions or analyses. All analyses were performed on wet
samples to ensure there were no losses of Hg(0). The moisture content of solid-phase samples was
determined on separate aliquots for each column by difference after drying at 35 °C and was 23 ±
2% ($n$ = 48) (Section S8).
THg and Hg stable isotope analyses were cold digested in modified aqua regia following the methods
described in McLagan et al. (2022) (1 mL nitric acid replace with 1 mL BrCl). Analyses of THg
concentrations from the digestion extracts were determined using CV-AAS following DIN method
1483 and USEPA method 1631. Results are reported on a dry weight basis and moisture content was
determined by difference after baking at 105 °C using aliquots of the solid-phase sample (Section
S8). Due to the low concentrations in the original solid-phase aquifer materials, THg concentrations
were measured with a DMA80 (Milestone SCI) via thermal decomposition, amalgamation, and AAS
(Table 1).



Speciation analyses were performed by pyrolytic thermal desorption (PTD), which continually
measures Hg at 254 nm within an AAS detector that is connected to a sample combustion furnace
that heats samples from room temperature to 650°C a 1°C per minute in a stream of $N_2$ gas. This
method is described in detail by Biester and Scholz (1996). The sample release curves were
compared to the release curves for a series of Hg reference materials (Hg(0), $HgCl_2$, $Hg_2Cl_2$ (calomel),
cinnabar: α-HgS, metacinnabar: β-HgS, and $Hg^{2+}$-sulphate: $HgSO_4$) in silicon dioxide ($SiO_2$) matrix
(see Section S9 for reference material curves) to qualitatively assess the species or "fractions" of Hg
present in the samples.
### 2.2.3    Liquid- and solid-phase Hg stable isotope analyses
Samples for stable Hg isotope analyses included stabilized liquid-phase eluate samples and solid-
phase aqua-regia extracts diluted with deionised water (18.2 MΩ cm). Liquid-phase samples were
collected in 15 mL polypropylene tubes and stabilized with BrCl to reach 1% of the sampled volume.
Analyses were made using a Nu Plasma II (Nu Instruments) multicollector inductively coupled
plasma mass spectrometer (MC-ICP-MS) with a cold-vapor generator (HGX-200; Teledyne Cetac)
that allows direct addition of Hg(0) into MC-ICP-MS plasma by reducing all Hg in samples with $SnCl_2$.
The isotope ratios were determined relative to NIST-3133 (National Institute of Standards and
Technology; NIST) using the standard bracketing approach and corrected for mass-bias using
thallium (Tl) doping from NIST-997 (NIST) introduced using an Aridus-2 desolvating nebulizer
(Teledyne CETAC). MDF was assessed by variation in $δ^{202}Hg$, while $Δ^{199}Hg$, $Δ^{200}Hg$, $Δ^{201}Hg$, and
$Δ^{204}Hg$ were used to assess MIF of odd and even isotopes) (see Grigg et al., 2018; McLagan et al.,
2022 for method details).
### 2.2.4    Complementary analyses
Metal cations in the solid- and liquid-phases were measured with inductively coupled plasma optical
emission spectrometry (ICP-OES; Varian 715-ES; Agilent Technologies Inc.). Solid-phase total carbon
(TC), total organic carbon (TOC), and total inorganic carbon (TIC; dissolved by hydrochloric acid)
were measured by infra-red detection of CO2 released (DIMA 1000NT; Dimatec, Germany).
Dissolved organic carbon of stock solution and eluate was measured with a carbon/nitrogen
analyser (Multi N / C 2100; Analytic Jena) (see Section S2). Liquid-phase dissolved oxygen content,
redox potential, electrical conductivity, and pH were measured by handheld probes.
### 2.2.5    Retardation ($R_D$) and sorption/partitioning/distribution ($K_D$) coefficient calculations
The retardation coefficient ($R_D$) is essentially the ratio of the velocity of the water front ($v_w$) and
velocity of the Hg front delayed by sorption processes ($v_{Hg}$) moving through the columns (Equation
1). Since the path of the the soluble pollutant (Hg) and water are the same, transport time can be
determined based on the time it takes the fronts to pass through the columns ($t_{Hg}$ and $t_w$,
respectively). NaCl breakthrough curve was used as a proxy for water based on the assumption it is
a conservative tracer. $t_{Hg}$ and $t_w$ are given when the respective ratios of the NaCl and THg
concentrations in the eluate is equal to half the input concentration (stock solution; $C_{eluate}$ / $C_{initial}$ =
0.5) (Patterson et al., 1993; Reichert, 1991; Schnaar and Brusseau, 2013).
$$R_D = {^{v_w}}/{_{v_{Hg}}} = {^{t_w}}/{_{t_{Hg}}}$$    Equation 1
$R_D$ is related to the sorption or partitioning or distribution coefficient ($K_D$; mL $g^{-1}$) according to
Equation 2 and Equation 3 (USEPA, 2004):

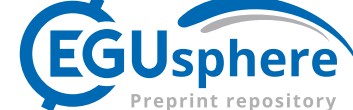

$$R_D = 1 + (\rho_b/n_e)K_D \hspace{6cm} \text{Equation 2}$$
$$K_D = (R_D - 1)(n_e/\rho_b) \hspace{6cm} \text{Equation 3}$$
Where, $n_e$ is the effective porosity (EXP1: 0.470 ± 0.008, $n$ = 3; EXP2: 0.459 ± 0.004, $n$ = 3), which is
the ratio of the column pore volume (EXP1: 23.3 ± 0.5 mL, $n$ = 3; EXP1: 22.5 ± 0.1 mL, $n$ = 3) to the
total volume of the solid-phase materials of the columns (EXP1: 49.7 ± 0.3 mL, $n$ = 3; EXP2: 49.0 ±
0.5 mL, $n$ = 3). $R_D$ could only be calculated for columns that went to equilibrium and desorption (not
50% breakthrough), $n_e$ was calculated for columns that went through desorption (C1.1-C1.3 and
C2.1-C2.3); and hence, $K_D$ was only calculated for these columns. Note, the pore volumes reported
above are the data used when reporting the number of pore volumes.

## 2.3 Quality Assurance and quality control (QAQC)

For liquid-phase analyses, a 140.8 ng L$^{-1}$ Hg(II) stock solution (Sigma Aldrich) was measured
throughout the analyses and recovery was 99 ± 5% ($n$ = 250). For solid-phase analyses, Chinese Soil
(NCS DC73030; Chinese National Analysis Centre for Iron and Steel) was measured and recovery was
101 ± 6% ($n$ = 16). The accuracy and precision of Hg stable isotope measurements was assessed
using the "in-house" *ETH Fluka* standard. Mean values across the measurement sessions were:
$\delta^{202}$Hg = -1.42 ± 0.08 ‰; $\Delta^{199}$Hg = 0.08 ± 0.02 ‰; $\Delta^{200}$Hg = 0.02 ± 0.02 ‰; $\Delta^{201}$Hg = 0.03 ± 0.03 ‰;
$\Delta^{204}$Hg = -0.01 ± 0.06 ‰ ($n$ = 26; all uncertainty values are reported as 2SD). All uncertainties are
1SD, unless otherwise reported (i.e., 2SD used to report Hg stable isotope analysis uncertainty.
These values are within the range of other studies (i.e., Obrist et al., 2017; Goix et al., 2019; McLagan
et al., 2022). Theoretical solid-phase THg concentration (compared to measured THg
concentrations) are determined via mass balance of liquid-phase THg concentrations of stock
solution and eluate and the volume of stock solution applied to the columns. All statistical tests and
sorption fitting comparisons were performed in OriginPro 2018 (Origin Lab Corporation).

# 3 Results and discussion

## 3.1 Sorption and desorption behaviour of mercury in column experiments

### 3.1.1 Sorption

As expected, the uptake of the HgCl$_2$ solution to the solid phase aquifer materials followed an S-
shaped breakthrough curve best described by the Freundlich model (Figure 2). Initially, >99.9% of
the Hg in solution was sorbed to the solid phase materials and 1.0-1.3 L (43 − 55 pore volumes) and
0.3-0.45 L (13 − 16 pore volumes), in EXP1 and EXP2, respectively, was required to reach eluate THg
concentrations equivalent to 1% of stock solution (Section S5). This was followed by a phase of rapid
increase in the eluate concentrations (decreasing fraction of the Hg in solution sorbing to the solid-
phase). Finally, the increase in eluate THg concentration slowed as it approached the upper
asymptotic bound of the original stock solution concentration in each experiment and equilibrium
of Hg fluxes between the solid- and liquid-phases was approached/reached. EXP1 likely did not
completely reach a stable equilibrium point (eluate concentration was at ≈91% of stock solution
concentration when the stock solution was changed to water), and more time/volume of solution
was required. This would have required creation of more stock solution; instead, green chemistry
prevailed, and the choice was made to move onto the desorption phase with consideration of the
higher concentration (faster) follow-up EXP2. This behaviour was similar to the one other detailed
study on Hg sorption in natural soils with sufficient liquid-phase sampling frequency to create





column breakthrough curves on OM-rich (9.4 – 24.7% OM) Amazonian soils and similar stock
solution concentrations (60 – 120 mg L$^{-1}$; Miretzky et al., 2005). Qualitative liquid-phase Hg
speciation analyses confirm that the majority of Hg was dissolved inorganic Hg(II) (EXP1: 83 ± 6%;
EXP2: 77 ± 8%), a fraction of which will be soluble HgCl$_2$ (species used generate stock solution), but
also fractions of hydrolysed species (i.e., HgClOH, Hg(OH)$_2$, [HgCl$_3$]$^-$) formed in solution at pH in the
observed range (7.7 – 8.1) of these experiments (Delnomdedieu et al., 1992; Gunneriusson and
Sjöberg, 1992; Kim et al., 2004; see also Section S10 for theoretical Hg speciation results using Visual
MINTEQ v3.1). These liquid-phase Hg speciation results are similar to those reported for
groundwater samples previously collected at the contaminated site where these materials were
extracted from (Bollen et al., 2008; Richard et al., 2016a; McLagan et al., 2022).

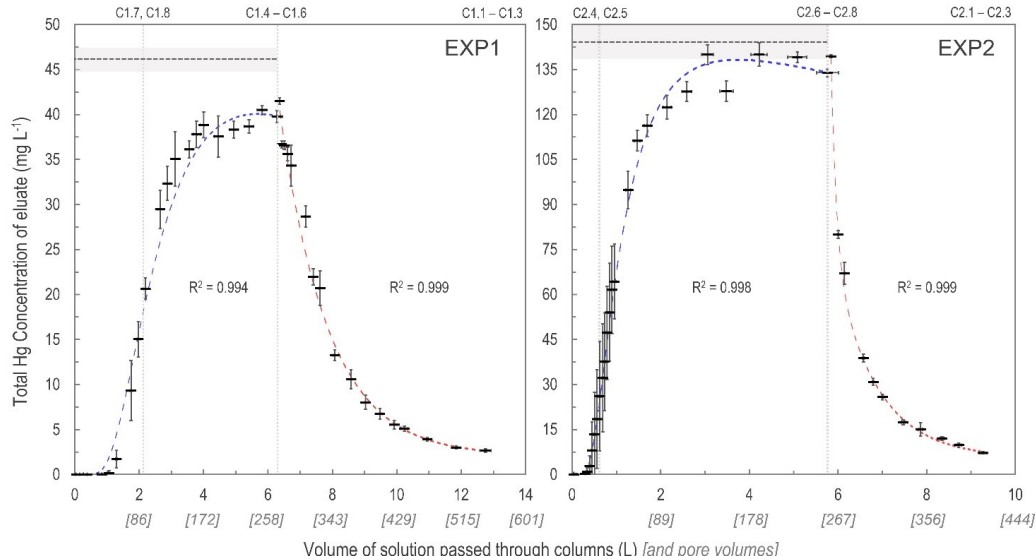


*Figure 2: Total Hg concentration eluate breakthrough curves for low (EXP1; left panel) and high (EXP2; right panel) concentration stock solution experiments. Horizontal dashed lines (mean) and shaded area (1SD) indicate the original stock solution concentrations in each experiment and vertical dotted lines indicate column removal points (column IDs above panels indicate which columns were removed). Uncertainty in the x-axis relates to the differing volumes passed through individual columns at each sampling period. Sorption curves were fitted with Freundlich functions (blue dashed lines), and desorption curves were fitted with exponential decay functions (red dashed lines). These relationships presented the best fits compared to the fit of sorption functions and full details of these functions are listed in Section S6.*

Despite the very low OM content (Table 1) within these solid-phase aquifer materials, the
equilibrium uptake capacity was very high in both experiments. These concentrations were
determined both (i) analytically by solid-phase THg analyses, and (ii) theoretically, based on the
inverse of the breakthrough curve integral: the area above the curve and below the stock solution
concentration. This has been referred to as "holdup" (*H*; mg of Hg), (Van Genuchten and Parker,
1984) and is described in Equation 4:
$H = [C_0 V_f - \int C_e \, dV]$                                                                                      Equation 4



Where, $C_e$ is the eluate THg concentration (mg L$^{-1}$), $C_0$ is the stock solution THg concentration (mg
L$^{-1}$), and $V_f$ is the accumulated solution volume that has passed through the columns at the point
they were removed (L). Theoretical concentrations reached 1880 ± 20 mg kg$^{-1}$ in EXP and 2810 ± 40
mg kg$^{-1}$ in EXP2 (Table 2; Section S3). These data are directly comparable, and indeed within the
same range as the theoretical solid-phase concentrations calculated by Miretzky et al. (2005) for the
OM-rich Amazonian soils (THg concentrations: 950 – 3960 mg kg$^{-1}$). The elevated Hg sorption
observed by Miretzky et al. (2005) is to be expected due to the affinity of Hg for OM (e.g., Yin et al.,
1996; Jiskra et al., 2015; Manceau and Nagy, 2019). Nonetheless, Miretzky et al. (2005) found their
calculated solid-phase THg concentrations at equilibrium (sorptive capacity of the soils) were
greater when OM% + clay% was considered rather than OM% alone was considered (Miretzky et al.,
2005), which highlights the potential role clay (and oxide) minerals can play in Hg sorption to solid-
phase soil or aquifer materials.
Hg sorption to OM has been observed to increase at lower pH (Andersson, 1979; Yin et al., 1996).
However, the opposite has been reported for sorption of Hg to clay minerals: in neutral and slightly
basic soils, the sorption capacity is controlled by the mineral components (Andersson, 1979;
Schuster, 1991; Gabriel and Williamson, 2004). Indeed, the pH range of the eluate and stock solution
(pH range: 7.7 – 8.1) present ideal conditions for Hg sorption to clay minerals and Fe and Mn
(oxy)hydroxide minerals. Hg sorption to these inorganic minerals becomes more likely in our
experiments considering the very low OM content of the solid-phase materials (Table 1). Haitzer et
al. (2002) estimated that at ratios of THg-to-OM above 1 µg of Hg per mg of OM the strong thiol-
group bonding sites for Hg within OM are saturated. Based on the TOC data of these solid-phase
materials (assuming 0.16% TOC = 0.32 % OM), there would be 224 mg of OM within a column. To
surpass the ratio of 1 µg of Hg per mg of OM, only 4.9 and 1.6 mL of stock solution or 0.21 and 0.07
pore volumes in EXP1 and EXP2, respectively, would need to be added to the columns to saturate
the strong thiol-group binding sites with Hg. Considering that Hg breakthrough occurred only after
about 50 and 15 pore volumes in EXP1 and EXP2, respectively, it can be assumed that not only the
strong Hg-binding thiol-groups but also the other less strong Hg-binding functional groups (e.g.,
carboxyl groups) of the small OM pool in the columns were fully saturated early in the experiments.
Hence, solid-phase sorption of Hg within these experiments was dominated by interactions with
inorganic minerals. The role of such inorganic minerals was also highlighted in one of the few studies
that exist examining Hg transport and fate in aquifers (Lamborg et al., 2013).
*Table 2: Theoretical (liquid-phase THg mass-balance) and measured solid-phase THg concentrations*
*and recovers of the measured-to-expected (theoretical) concentrations for each the columns in EXP1*
*and EXP2.*

| Experiment 1 (EXP1; 46.1 ± 1.1 mg L$^{-1}$) | | | | | Experiment 2 (EXP2; 144 ± 6 mg L$^{-1}$) | | | | |
|---|---|---|---|---|---|---|---|---|---|
| Column | Stage | Theoretical Hg conc. (mg kg$^{-1}$) | Measured Hg conc. (mg kg$^{-1}$) | Recovery | Column | Stage | Theoretical Hg conc. (mg kg$^{-1}$) | Measured Hg conc. (mg kg$^{-1}$) | Recovery |
| C1.1 | Desorption | 820 | 722 ± 91 | 88.0% | C2.1 | Desorption | 1360 | 1060 ± 230 | 78.3% |
| C1.2 | Desorption | 890 | 877 ± 206 | 98.6% | C2.2 | Desorption | 1300 | 786 ± 390 | 60.2% |
| C1.3 | Desorption | 847 | 835 ± 120 | 98.6% | C2.3 | Desorption | 1490 | 1050 ± 57 | 70.1% |
| C1.4 | Equilibrium | 1870 | 1470 ± 221 | 78.5% | C2.4 | 50% breakthrough | 1030 | 785 ± 220 | 76.1% |
| C1.5 | Equilibrium | 1910 | 1630 ± 286 | 85.1% | C2.5 | 50% breakthrough | 1140 | 702 ± 330 | 61.4% |
| C1.6 | Equilibrium | 1870 | 1440 ± 92 | 77.1% | C2.6 | Equilibrium | 2770 | 2380 ± 452 | 86.1% |
| C1.7 | 50% breakthrough | 1320 | 1470 ± 384 | 111.3% | C2.7 | Equilibrium | 2850 | 2320 ± 388 | 81.2% |
| C1.8 | 50% breakthrough | 1300 | 960 ± 524 | 73.6% | C2.8 | Equilibrium | 2820 | 2260 ± 272 | 79.8% |



Measured THg concentrations were typically lower than the theoretical calculated values (Table 2)
and contaminant masses can be difficult to balance in contaminant batch and column experiments
(Van Genuchten and Parker, 1984; Hebig et al., 2014). This is of particular concern for a contaminant
such as Hg whose stability and contamination issues have been widely studied due to the capacity
of different Hg species to sorb to and diffuse through plastic polymers (at differing rates) (Hall et al.,
2002; Parker and Bloom, 2005; Hammerschmidt et al., 2011). Loss of a fraction of the THg in solution
to/through tubing and the walls of the column is likely contributing to the lower recovery in some
of these samples. Other factors that could be contributing to the differences between the
theoretical and measured concentrations are heterogeneity of the solid-phase and solid-phase
sample extraction (particularly during movement of the Hg mass transfer front), loss of Hg from
solid-phase before sample extraction and analyses (particularly for volatile Hg(0); Parker and Bloom,
2005), and inherent analytical uncertainties The heterogeneity of the materials is emphasized by
the absence of trends in THg concentrations within the sections of the columns, even for the
columns undergoing movement of the mass transfer zone (see Section S8). Unfortunately, Miretzky
et al. (2005) did not provide total sampling volumes for their experiments and no assessment of
measured THg recoveries was (or can be) made for direct comparison to our recovery data.

### 3.1.2   Desorption

The desorption phase of both EXP1 and EXP2 followed an exponential decay model; results confirm
that sorption is (partially) reversible and initially rapid (Figure 2). After the stock solution was
switched to water for the desorption phase, the eluate solution reached <50% of the stock solution
THg concentration with additions of ≈1 L (≈43 pore volumes) and ≈0.5 L (≈22 pore volumes) of
solution in EXP1 and EXP2, respectively (Figure 2). At the termination of the experiments eluate THg
concentrations dropped to <10% of the original stock solution (Figure 2). While it is evident that
more Hg would have been released if desorption was permitted to proceed further (terminated due
to time and to prevent excess contaminated waste solution), measured data indicated that 46 ± 6%
(Theoretical: 55 ± 2%) in EXP1 and 58 ± 10% (Theoretical: 51 ± 4%) in EXP2 of THg could be extracted
from the solid-phase materials before the experiments were terminated. Evidence from the
contaminated aquifer where these solid-phase materials were extracted suggest that the retention
of a fraction of this Hg within the solid-phase materials is long-term (Bollen et al., 2008; McLagan et
al., 2022). McLagan et al. (2022) report that elevated solid- (up to 562 mg kg$^{-1}$) and liquid-phase (164
± 75.4 μg L$^{-1}$) THg concentrations are still found at the site to the present day, more than 55 years
since the industrial use of Hg (kyanisation) at the site ceased.
The authors of that study associate this residual retention of Hg to the diffusion of Hg into the
mineral matrix or secondary transformation to a more stable (and less soluble) Hg(II) species
(McLagan et al., 2022). Previous work agrees that sorption and subsequent release of Hg to/from
solid-phase soils and solid-phase materials is likely controlled by multiple processes (Yin et al., 1997;
Bradl, 2004; Reis et al., 2016). The more easily extractable Hg is likely to be associated with Fe and
Mn (oxy)hydroxide, and clay minerals through outer-sphere complexes that form through cation
exchange and electrostatic intermolecular forces (Bradl, 2004; Reis et al., 2016). Overtime, some of
the Hg associated through these weaker surface interactions will diffuse into the matrix and/or form
inner-sphere complexes, processes that both slow the release of the sorbed Hg (Bradl, 2004; Reis et
al., 2016). Similar results were observed by Miretzky et al. (2005) in the OM rich Amazonian soil
columns with 27 - 38% of Hg sorbed to the solid-phase materials being rapidly redissolved in the
initial desorption phase. However, the soils with higher OM content showed stronger hysteresis and





considerably less Hg was released during the second phase of desorption (Miretzky et al., 2005) than
in our low OM solid-phase materials suggesting stronger interactions of inner-sphere complexed Hg
with OM; results supported by work done in other studies examining Hg sorption to solid-phase
materials (Yin et al., 1996; Reis et al., 2016).

### 3.1.3 Insights from stable Hg isotopes

Variations in $\delta^{202}$Hg values, describing MDF of Hg isotopes, were observed in both the liquid- and
solid-phase across the experiments (Figure 3; Section S7; Section S8). During the initial phase of the
experiments (before eluate breakthrough), transfer of Hg from the applied stock solution
($\delta^{202}$Hg: -0.61 ± 0.01‰ relative to NIST-3133, 1SD; $n$ = 3) to the solid-phase materials is complete.
When there is complete transfer of a "pool" of Hg from reactants to products there is complete
transfer of stable isotopes; and hence no fractionation can be observed.

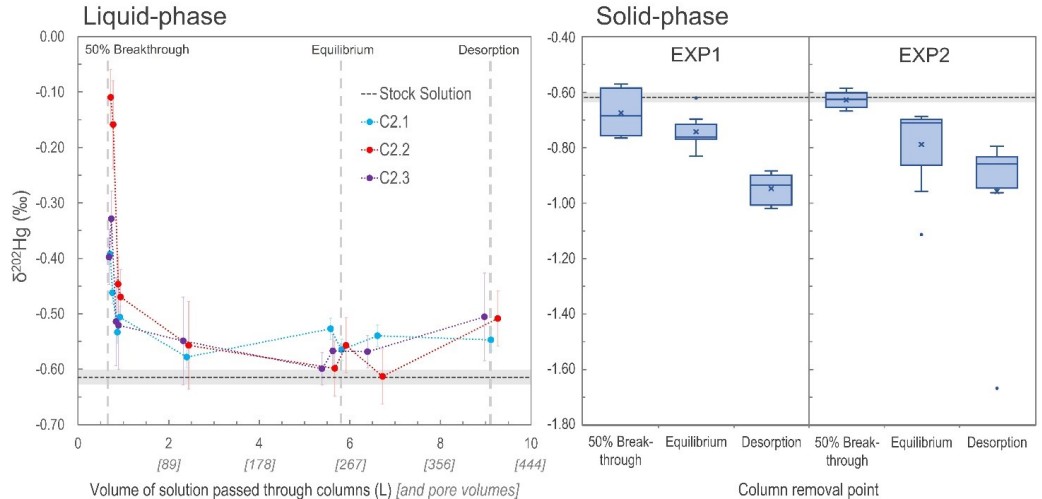


*Figure 3: Development of liquid-phase $\delta^{202}$Hg values for columns C2.1 – C2.3 measured at nine*
*intervals during EXP2 (left panel), and box plots of solid-phase $\delta^{202}$Hg values measured in both EXP1*
*and EXP2 ("x" denotes mean values, dots denote outliers). In both panels, the grey dash line*
*represents the mean $\delta^{202}$Hg value (light grey rectangle: 1SD) measured for the stock solution. Note,*
*the vertical grey dashed lines indicating solid-phase column removal points in the left panel are only*
*approximations as the liquid-phase stable isotope measurements were only made on columns C2.1-*
*2.3 that proceeded until the end of desorption.*

Once Hg begins to breakthrough the columns, the eluate is initially enriched in heavy isotopes
associated with the preferential transfer (sorption) of lighter isotopes to the solid-phase materials
(Jiskra et al., 2012; Wiederhold, 2015) with heavier isotopes retained in solution and passed into the
eluate. In all three of the EXP2 columns examined for stable isotopes in the liquid-phase, the first
two liquid-phase stable isotope samples (sampled just after ≈50% breakthrough column removals)
had more positive $\delta^{202}$Hg values than the remaining liquid-phase samples (Figure 3). However, it is
also apparent that at ≈50% breakthrough, there was little MDF imparted on the solid-phase
materials compared to the stock solution (Figure 3). This ostensibly contrasting finding (observable
positive MDF in the liquid-phase and little negative MDF in the solid-phase) can be explained by the
proportion of Hg transferred to the solid-phase of the total mass added in solution. At the 50%
breakthrough column removal, the proportion of Hg sorbed by the columns was 95.4 and 90.4%,





respectively for C1.7 and C1.8 (EXP1) and 83.8 and 88.5%, respectively for C2.4 and C2.5 (EXP2;
based on theoretical calculations). The majority of this sorption occurred during the complete (or
near-complete) transfer of isotopes before (or just after) eluate breakthrough. Hence, the MDF that
began to occur after breakthrough (observable in the early liquid-phase eluate samples) had little
influence on the Hg stable isotope ratios of the solid-phase materials of columns removed at the
≈50% breakthrough point.
This process is further supported when examining the $\delta^{202}$Hg values of the column layers at ≈50%
breakthrough. The bottom layers of C1.7 ($\delta^{202}$Hg: -0.76 ± 0.07‰) and C1.8 ($\delta^{202}$Hg: -0.75 ± 0.07‰)
in EXP1 were more negative than the stock solution, while the top layers ($\delta^{202}$Hg: -0.57 ± 0.15‰ and
$\delta^{202}$Hg: -0.59 ± 0.07‰ for C1.7 and C1.8, respectively) were equivalent to the stock solution (Section
S8). These data suggest observable MDF was beginning to occur in the part of the column exposed
to the Hg front (bottom) for the longest. The same was not the case in EXP2 (no observable trend in
$\delta^{202}$Hg between layers; Section S8). We attribute this to the more elevated THg concentrations and
faster movement of the Hg front moving through the columns (see Table 3 below) in EXP2
overwhelming the layering MDF observed in EXP1.
As sorption progresses to equilibrium, we observe a negative shift in the eluate $\delta^{202}$Hg value of all
three columns falling in the range of ≈-0.6 to -0.5‰, which is slightly more positive than the stock
solution ($\delta^{202}$Hg: -0.61 ± 0.01 ‰ 1SD; ± 0.08 ‰ analytical 2SD; Figure 3). During this transition in the
Hg uptake process the net effect is that most, and then essentially all, Hg input from the stock
solution is passing through the columns and into the eluate and any kinetic MDF occurring would
be limited. Nonetheless, equilibrium-based isotope exchange would also drive lighter isotopes into
the solid-phase materials (Wiederhold et al., 2010; Jiskra et al., 2012; Wiederhold, 2015), which is
the likely explanation for the liquid-phase $\delta^{202}$Hg values remaining slightly more positive than the
stock solution. While the impact of this MDF on the continuously flowing eluate is small when the
system is at equilibrium, the effect of this equilibrium-based MDF on the solid-phase is more
manifest as its effect is cumulative. Overtime, more and more lighter isotopes preferentially sorb to
the solid-phase; and hence, the mean $\delta^{202}$Hg values of the solid-phase materials in EXP1
($\delta^{202}$Hg: -0.74 ± 0.06‰ 1SD) and EXP2 ($\delta^{202}$Hg: -0.79 ± 0.15‰ 1SD) at the end of the sorption
experiments (at or near column equilibrium) are more negative than the stock solution (and solid-
phase materials at ≈50% breakthrough). Thus, we suggest equilibrium-based MDF (with some
potential for kinetic MDF contributions) to be the primary driver of the more negative $\delta^{202}$Hg values
observed in the solid-phase materials at the end of the equilibrium-phase of the experiments. These
observations agree with the observed results of McLagan et al. (2022) sampled within the
contaminated aquifer adjacent to which these uncontaminated materials were derived.
At the end of the desorption phase, the solid-phase materials have undergone further MDF to more
negative $\delta^{202}$Hg values (EXP1 $\delta^{202}$Hg: -0.95 ± 0.05‰; EXP2 $\delta^{202}$Hg: -0.96 ± 0.27‰ 1SD). Two of the
three columns monitored for liquid-phase stable isotopes at the end of desorption also show a slight
positive MDF shift and values for all three columns are slightly more positive ($\delta^{202}$Hg: -0.55 to -0.51
‰) than the stock solution (Figure 3). As discussed, desorption proceeds via a two-step mechanism:
a rapid initial desorption as easily exchangeable, outer-sphere complexed Hg is released, followed
by a slower phase of desorption as this easily exchangeable pool depletes. Brocza et al. (2019) and
McLagan et al. (2022) suggest that this easily exchangeable pool is enriched in heavier isotopes
compared to the fraction that diffuses into the mineral matrix or transforms to more stable, less
soluble Hg(II) species as these secondary processes favour lighter isotopes. Thus, removal of the





heavy isotope enriched, easily exchangeable pool of Hg is the likely driver of more negative $\delta^{202}$Hg
values in the solid-phase materials after desorption. While Demers et al. (2018) studied
predominantly surface water samples linked to Hg soil-groundwater contamination at a site in
Tennessee, USA (industrial use of Hg(0)), they did observe more positive $\delta^{202}$Hg values with elevated
dissolved THg concentrations values in samples from the hyporheic zone associated with exfiltrating
groundwater from the contaminated areas. These data would agree with the more positive liquid-
phase $\delta^{202}$Hg values observed in our study and by McLagan et al. (2022).
Variation in both odd- and even-isotope MIF was within the range of analytical uncertainties
(Section S7; Section S8). McLagan et al. (2022) did observe small variation in $\Delta^{199}$Hg between solid-
and liquid-phases, which the authors suggest may be linked to MIF driven by dark abiotic reduction
of Hg(II) (Zheng and Hintelmann, 2010). However, it is unlikely that this process could manifest into
an observable change in $\Delta^{199}$Hg considering the short duration of these experiments even if the
process could occur at all within these columns.

## 3.2   Is reduction of Hg(II) to Hg(0) occurring within the columns?

Reduction of Hg(II) to Hg(0) has been observed previously at this and other sites impacted by
kyanisation activities (Bollen et al., 2008; Richard et al., 2016a; 2016b; McLagan et al., 2022). In
these subsurface environments with low OM and very high THg concentrations, this secondary Hg(0)
production has been linked to abiotic, (hydr)oxide mineral surface catalysed reactions driven by
other redox active metals (Bollen et al., 2008; Richard et al., 2016a; 2016b). Since $HgCl_2$ solution was
the only form of Hg applied in the column experiments, the presence of Hg(0) in either the liquid-
or solid-phases must be explained via reduction of Hg(II).
To examine the presence of Hg(0), PTD analyses were run on the (undried) solid-phase materials
from the columns after the sorption experiments. The PTD extinction curves showed little variation
across all sections of all columns from either experiment (see Section S9). All curves mimic the low
sample weight (≈0.1 g) mean extinction curves displayed in Figure 4 and are dominated by a single
peak with a maximum release of ≈225 °C, which aligns with the maximum extinction of the $HgCl_2$
standard in silicon dioxide ($SiO_2$). This supports the hypothesis of direct (outer-sphere) complexation
or electrostatic interaction of dissolved Hg(II) species to the mineral surfaces posited previously
(Bradl, 2004; Reis et al., 2016) and by McLagan et al. (2022). Nonetheless, these low sample weight
PTD curves were indicative of some qualitative evidence of very small peaks at <175 °C; peaks in this
range are associated with Hg(0) (Biester and Scholz, 1996; McLagan et al., 2022). The initial sample
masses used in the PTD analyses were low (≈0.1 g) so as to not overwhelm the AAS detector, release
large amounts of gas-phase Hg(0), and potentially cause memory effects in future analyses.
Nevertheless, this would not occur if sample masses were increased (≈2.0 g) and the temperature
ramp stopped at ≈175 °C. When the solid-phase materials were analysed in this manner, Hg(0) peaks
were detected across all sections of all columns in both experiments (see Section S9; Figure 4).
Additionally, detectable concentrations of Hg(0) were observed across all of the qualitative liquid-
phase Hg speciation analyses and elevated above the Hg(0) concentrations measured in the stock
solution (Section S4). The observed liquid-phase fraction of Hg(0) was highest at the ≈25%
breakthrough sample collection point in EXP1 (0.7%) and EXP2 (0.1%) with the fraction being ≤0.1%
in all other samples (Section S4). While these data suggest that reduction of Hg(II) to Hg(0) begins
almost immediately after the introduction of the $HgCl_2$ solution, we link the declining proportion of



Hg(0) to the low solubility of Hg(0) (≈50 µg L$^{-1}$) (Skyllberg, 2012; Brocza et al., 2019), which was
already reached at the ≈25% breakthrough sample collection point in both experiments.

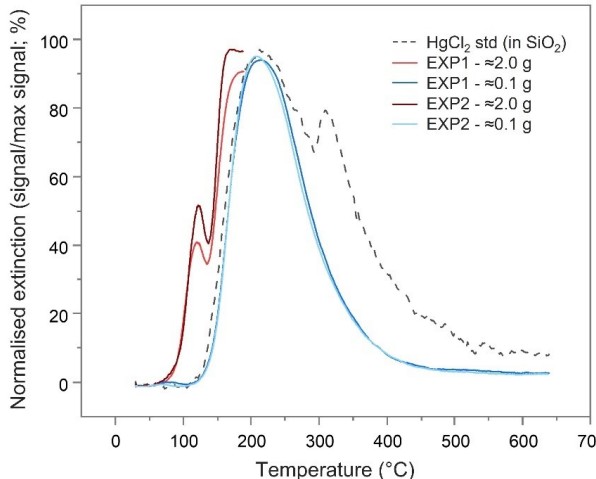


*Figure 4: Mean pyrolytic thermal desorption (PTD) extinction curves from solid-phase materials from*
*EXP1 and EXP2 assessed with two different sample masses. Analyses of the larger sample mass (≈2.0*
*g of material) were terminated when the temperature ramp reached ≈175 °C to prevent excessive*
*gas-phase Hg release and potential memory effects on the instrument.*

These measured Hg(0) fractions in solid- and liquid-phase analyses provide further direct evidence
of Hg(0) production under saturated, oxic conditions in low OM solid-phase materials. Hg(0)
production in these contaminated aquifers has been linked to the slower than expected horizontal
progress of the plume of Hg in the aquifer at the site where this contamination occurred (Bollen et
al., 2008; Richard et al., 2016a; 2016b; McLagan et al., 2022). While these data indicate that fraction
of Hg(0) produced is relatively small, the volume of soil and aquifer materials in which this process
can occur is large. The contamination plume of the aquifer at the site where the solid phase
materials were removed from is ≈1000 m with an area of ≈6x10$^4$ m$^2$ (Bollen et al., 2008; McLagan
et al., 2022). If we conservatively assume a mean depth of contamination of 2 m, mean THg
concentration of 2 mg kg$^{-1}$, the fraction of Hg(0) produced per day is 0.01 – 0.001% of the THg (all
conservative estimates based off data from Bollen et al., 2008; McLagan et al., 2022), and the same
bulk density and flow rates as in our experiments, we can produce a *back-of-the-envelope* estimate
of the mass of Hg(0) produced and potentially lost from the aquifer to overlying soils. Based off
these numbers, we estimate that 0.3 – 0.4 g of Hg(II) is transformed to Hg(0) each day within the
aquifer of the contaminated site in southern Germany; over the course of one-year, this equates to
the transformation ≈5 – 15 kg of Hg(II) to Hg(0). Even a relatively conservative estimate of the
conversion (and potential loss) of this mass of Hg(II) in contaminated aquifers such as this provides
strong evidence that the process of Hg(II) reduction plays a key role in limiting the transport of the
10-20 tonnes of Hg that was added to this soil-groundwater system in the ≈120 years since industrial
operations commenced.
## 3.3   Retardation (R$_D$) and sorption coefficient (K$_D$) calculations
As expected, R$_D$ values were substantially greater than 1, confirming substantial interaction
between the applied HgCl$_2$ solution and the solid-phase aquifer materials (Table 3). The difference



in $R_D$ and $K_D$ values between EXP1 and EXP2 (Table 3) indicate stock solution concentration is a factor
in the transport of mercury within these columns. The elevated stock solution concentrations may
be undermining the assumption of equal accessibility to sorption sites (USEPA, 2004). However, the
purpose of these experiments was to simulate the original contamination by the industrial
use/misuse of $HgCl_2$ solution, and while we can only estimate original concentration of solution
being transported through the soil-groundwater system, we do expect they were very high due to
the extent (both in terms of elevated concentrations and three-dimensional spread of the
contamination plume) of contamination that remains and the very high concentration of the
solution used in rot-prevention treatment of timber (Bollen et al., 2008; Richard et al., 2016a;
McLagan et al., 2022). Considering the high concentrations of Hg that have been observed within
this and other Hg contaminated aquifers (Katsenovich et al., 2010; Lamborg et al., 2013; Demers et
al., 2018) it is critical that we do not isolate our study of Hg transport dynamics to low concentration
experiments that meet assumptions for theoretical sorption (batch and column) experiments.
*Table 3: Calculated retardation ($R_D$) and sorption ($K_D$) coefficients for EXP1 and EXP2 (definitions are*
*given in Section 2.2.5).*

| EXP1 | | | | | EXP2 | | | | |
|---|---|---|---|---|---|---|---|---|---|
| Column | $t_w$ (min) | $t_{Hg}$ (min) | $R_D$ | $K_D$ (mL g$^{-1}$) | Column | $t_w$ (min) | $t_{Hg}$ (min) | $R_D$ | $K_D$ (mL g$^{-1}$) |
| C1.1 | 48.9 | 3628 | 74.7 | 23.8 | C2.1 | 43.0 | 1615 | 37.6 | 11.8 |
| C1.2 | 41.0 | 3629 | 88.5 | 29.5 | C2.2 | 38.2 | 1567 | 41.2 | 12.9 |
| C1.3 | 50.0 | 3779 | 75.6 | 25.1 | C2.3 | 45.8 | 1837 | 39.9 | 12.6 |
| C1.4 | 49.5 | 3678 | 74.3 | - | C2.6 | 41.0 | 1438 | 35.1 | - |
| C1.5 | 44.0 | 3488 | 79.3 | - | C2.7 | 44.1 | 1623 | 36.9 | - |
| C1.6 | 47.8 | 3599 | 75.3 | - | C2.8 | 37.5 | 1317 | 35.1 | - |
| | | **Mean** | **77.9** | **26.1** | | | **Mean** | **38.4** | **12.4** |
| | | **SD** | **5.5** | **3.0** | | | **SD** | **2.7** | **0.6** |


$R_D$ values can be calculated from Miretzky et al. (2005) based on the inverse of their v/v$_{water}$ value
and the mean of these derived $R_D$ values is 48 ± 13 for the high OM Amazonian soils. This again
affirms the high sorptive capacity of our low OM solid-phase aquifer materials at these comparative
concentration $HgCl_2$ applications. Lamborg et al. (2013) calculated $K_D$ values for a Hg contaminated
(from wastewater treatment) aquifer between 100 and 6300 mL g$^{-1}$ (log $K_D$: 2-3.8); yet calculations
had to assume liquid-phase concentrations from other studies. Log $K_D$ values calculated from soil
and sediment batch experiments typically range from ≈2 in lower OM materials (Akcay et al., 1996)
up to ≈6 in higher OM materials (Lyon et al., 1997). The logical next step is to utilise the measured
$R_D$ and $K_D$ data from our study to perform soil-groundwater modelling to better understand Hg
transport in this and other soil-groundwater systems as there are no previous estimates of $R_D$ and
$K_D$ values based on measured data for low OM solid-phase aquifer materials. The range of coefficient
values from ours and other studies described above relating to differing solid-phase properties,
input solution speciation, and assumptions used highlights the caution that should be made applying
these values to other systems as $R_D$ and $K_D$ values tend to be highly site specific (USEPA, 2004).

## Acknowledgements

We would like to thank Adelina Calean and Petra Schmidt for their support and contributions in
terms of experimental setup and sample analyses (including A.C. travelling to Vienna for to assist



with isotope analyses). We also thank undergraduate students Jan Pietrucha, Jette Greiser, and
Katja Braun for helping with liquid-phase sample collection and analyses. We thank Stephan M.
Kraemer for supporting the Hg isotope analyses at the University of Vienna. We would also like to
acknowledge Thomas Schöndorf from HPC Environmental Consulting for providing the solid-phase
materials used in this study. Also thanks to Hans Esser for helping design the eight-column holding
rack used in the experiments. This research was funded by the German Science Foundation (DFG)
grant BI 734/17-1 to H.B. and the Austrian Science Fund (FWF) grant I-3489-N28 to J.W. D.S.M.
would like to thank for support provided through a National Sciences and Engineering Research
Council of Canada (NSERC) postdoctoral fellowship.

## Author contributions

D.S.M., C.E., and H.B. designed the study and experiments with some feedback from other co-
authors, particularly J.-H.R during preliminary experiments. C.E. led all concentration and speciation
analyses with assistance from D.S.M. Isotope analyses were led by L.S. with assistance from J.W.
(and A.C. see above). This work was the basis for C.E.'s master's thesis, which was written in German.
The manuscript first draft was written by D.S.M. and all other authors provided feedback in building
the manuscript towards submission. Figures, tables, and SI were produced by D.S.M, C.E., and L.S.

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
