# Peer review of "Organic matters, but inorganic matters too: column examination"

_EGUsphere, 2023_

## Author Comment (AC1)

*We use the following format for comments and responses:*

*R1C1 = Reviewer 1, comment 1. R1AR1 = Reviewer 1, Author Response 1.*

*General comments*

R1C1 - The manuscript by McLagan *et al.* is well-written and describes a well-constructed study. The experiments detailed in this manuscript are well documented, frequently sampled, and analyzed to answer important research questions for Hg contamination, transport, and sorption/desorption. The resulting data is valuable and provides noteworthy contributions for future Hg research to build on. The high sorptive capacity of Hg in these sediments is an exciting find. While data on Hg sorption and desorption and presented retardation and distribution coefficients for Hg at contaminated concentrations in environments with low DOM may be the main selling points of the study, the data on Hg isotope fractionation associated with processes of sorption and desorption and evidence of Hg reduction in oxygenated soils are also noteworthy take-aways. The manuscript is recommended for publication in SOIL with minor suggested edits.

R1AR1 – We thank the reviewer for their time and overall positive support of the manuscript.

R1C2 - A general suggestion for improvement relates to the possible comparisons between the experiments and the real-world sediments sampled for this study. It is detailed that 74% of the material was discarded during the sieving process (Ln 130-131). In addition, there is a mention of an "optimal density" for the column packing (Ln 147), and the Authors went through several packing methods in the preliminary experimental phase (S1) where the density appears to have been a major factor in the consideration. This raises the following questions: 1) what the target density is, and how does it relate to the real world, e.g., what system the experiments now can be said to most closely mimic, and 2) what the effect of sieving off ¾ of the total weight could have for translating the findings to real-world systems. Discussing this would be a welcome inclusion, as would a site description of where the samples were collected.

R1AR2 – We would address the reviewers concern about the removal of coarser (large size fraction) materials by highlighting the importance of surface interactions during sorption of analyte from liquid-to-solid phase. The "specific surface area", which relates to the surface area (SA)-to-volume (V) ratio of the materials is critical; fine materials have much greater SA:V ratios; and hence, fine materials dominate sorption. Coarser materials dominate mass but play only a minor role in analyte sorption; thus, they are removed from column experiments. [References: e.g. Zhu et al., 1997; Cucarella & Renman, 2009]. In addition, large materials create more heterogeneous and less reproducible columns, which also supports their removal

(Lewis & Sjöstrom, 2010). Hence, this size fraction selection (<2 mm) followed general practices in the column experiment literature.

In terms of column density, we followed guidelines presented by Lewis & Sjöstrom (2010) for these aquifer materials which have been defined as sandy-gravel. The following sentence has been added to the methods:

"According to Lewis and Sjöstrom (2010), the average bulk densities range from 1.2 – 2.0 g/cm3 for sands and 1.6 – 2.0 for gravel. Thus, we deem the achieved bulk density of the columns to be appropriate for these materials, particularly as densities of the removed coarse materials are higher (solid densities are estimated at 2.65 g cm3; Lewis and Sjöstrom (2010))."

R1C3 - It is also suggested that the text be edited in a few instances to improve the conciseness and structure. This includes the beginning of the introduction, which starts very broadly with little direct relevance to the study, and the section between Ln 462-475, which can be shortened to a few sentences to keep the spotlight on the major findings in the manuscript.

R1AR3 – [comment from leading author] I thank the reviewer for their suggestion (and this is similar to a suggestion made by reviewer 2). However, I deem this to be more related to preference of writing style. I prefer to introduce my papers with something different that provides unique information that some readers may learn and benefit from rather than repeating the same information that is stated in most every paper on a particular topic. For instance, I would ask how many papers has the reviewer read on mercury that have included some form of "mercury is a persistent, bioaccumulative, and toxic contaminant…" (or some other very well noted anecdote on mercury) within the first few sentences? Here, I have introduced information on why mercury is such a unique metal (its quantum/electronic structure), information that is not well known even to many mercury scientists.

Moreover, I would argue that this has much "direct relevance to the study" as the behaviour of mercury in such saturated environments is influenced greatly by the quantum/electronic structure. For instance, the volatility of elemental mercury, which we link as a potential loss process of Hg from groundwater (and surface soils and water) is directly linked to elemental mercury not forming metal-metal bonds. It is my stylistic preference that the introduction remain in this form; I do feel strongly that individual writing styles still very much have a space and a place within scientific writing.

With reference to Ln 462-475. This was discussed at length between co-authors as is not a straightforward concept. As such, this paragraph was very carefully worded to ensure all readers could follow the explanation. We are very encouraged that the reviewer understands this behaviour, but it may not be as straightforward for other

readers. We wish not to shorten the discussion to ensure that all readers can follow it clearly.

*Specific Comments*

R1C4 - Ln 172-174: The tested HgCl2 concentrations are presented as estimates of original concentrations during the years of kyanization activity despite being three magnitudes below this concentration level. It is clear that this is a rough estimate and that lower concentrations of stock solutions were not feasible due to experimental constraints. However, can any data or calculations be referenced to support estimating a 1000x concentration loss in the porewater over 5-6 decades?

R1AR4 - The original concentration of the solution applied during the kyanisation process was 0.66% or 6600 ppm HgCl2 solution. Contamination at the site was from spills of this solution, which contains the very soluble HgCl2. Thus, we know solution ~50x stronger the EXP2 and >100x stronger than EXP1 were entering the top of the soil profile and this site operated from 1904-1965 with 10-20 T of Hg being lost to the soils and aquifer. These data are stated in the study and other referenced work examining this site. Without question, experimental constraints (time) was a major factor in selecting these experimental stock solution THg concentrations, but considering the history of the site we do deem them to be applicable to this site and others. This is supported by the fact that Miretzky et al., 2005 (the only other study Hg column experiments) applied similar concentration to a similar column experiment design. This paragraph has been updated as follows to better describe these considerations:

"Stock solutions were 46.1 ± 0.1 mg L-1 in EXP1 (n = 6) and 144 ± 6 mg L-1 in EXP2 (n = 12) and were selected for (i) time considerations (see Figure S1.6) and (ii) as these values balanced HgCl2 concentration applied during industrial activities (6600 ppm; spillages of this solution to the top of the soil profile) and as estimates of the original concentrations of HgCl2 contaminated solution entering the soil-groundwater system considering recently measured groundwater concentrations up to 164 ± 75.4 µg L-1 are still observed 55 years after closure cessation of the industrial activities at the site the solid-phase materials were extracted (McLagan et al., 2022)."

R1C5 - Ln 181, Table S2.2: Based on the 24 and 48-hour analyses, a decision was made to let the columns equilibrate for seven days before starting the experiment. The analysis shows that more than two days are needed to achieve equilibration. Is there any data to back up that seven days should be enough?

R1AR5 – Equilibrating the columns with the uncontaminated solution for 7 days was deemed a conservatively long period of time to ensure the equilibration between solution and solid phase based on the 24 and 48 hour assessments. We have provided all data both related to the preliminary experiments and the actual

experiments. This includes a very detailed summary of the preliminary experiments to ensure readers could follow our experimental design process and more easily replicate if desired. We do not have data confirming complete equilibration at 7 days (it would have been included if it was available). However, with the extended equilibration period, all co-authors deemed this would not have a significant impact on the results, an estimate that we stand by now.

R1C6 - Ln 242: How can the Authors know that no Hg0 loss occurs when analysis is performed on wet samples? A reference would be appreciated.

R1AR6 – This is a good pick up by the reviewer and an oversight on our part. This will be changed to: "All analyses were performed on wet samples to minimise any potential losses of Hg(0)." Of course this relates to the fact that if samples were dried (oven, air, or freeze drying) much greater Hg(0) losses can be expected due the volatility of Hg(0), which is in turn linked to its quantum/electronic structure.

R1C7 - Ln 320-330, Figure 2: With the eluate concentration only reaching 91% of the stock solution concentration, the Authors state that EXP1 likely did not reach equilibrium. Despite this, the Freundlich model fit to EXP1 indicates that the maximum totHg eluate concentration was reached when ca 5.7 L solution had passed through, after which the increase in Hg in the eluate ceased. If EXP1 did not reach equilibrium, would a maximum eluate concentration be passed? Or, is there reason to doubt the appropriateness of the model for the sorption phase in EXP1?

R1AR7 – The model is fitted to the data, if we were able to extend the equilibration period further, the model would be different, but it is highly unlikely a different model (non-Freundlich model: i.e., Langmuir) would have fit the data better. Other data fitting models were applied to both the sorption and desorption components of both experiments, but in both cases Freundlich and exponential decay functions, respectively, were easily the best fits. Moreover, Freundlich functions are common in describing column experiment breakthrough curves (e.g., Pan et al., 2005).

R1C8 - Ln 557: How can it be known that Hg0 exceeded solubility after the 25% breakthrough if Hg0 was only qualitatively measured? If this is a speculation, that should be indicated more clearly.

R1AR8 – This is an excellent point. We will use the term "semi-quantitative" in place of qualitative to assess the liquid phase speciation analysis and make this change throughout. While these data have a high uncertainty, Hg(0) is the species that we have the most confidence in, particularly in the case of these experiments. Hg(0) is measured simply by purging the sample with nitrogen gas directly into the CVAAS (without any addition of BrCl). In these experiments, the CVAAS was in the adjacent room to the room where the column experiments were conducted. All Hg(0) analyses were performed immediately (<30 mins) from sample collection. Thus, we have a higher confidence in these Hg(0) results than we had from the McLagan et

al., 2022 paper that required fresh (unstabilised) samples to be transported from the site (across Germany) and up to 5 days could pass resulting in potential loss of Hg(0). The following sentence will be added to the methods:

"Confidence in liquid phase Hg(0) concentrations is higher than for other species, as these result from purging untreated samples of Hg(0) with nitrogen gas directly into the CVAAS. All Hg(0) samples were analysed within 30 mins of sample collection."

R1C9 - Ln 571-575: While I support the conclusions, the calculations for the Hg0 production in the contaminated sediments hinge on the assumption that the experiments will mimic what is happening in the contaminated site sediments. Several factors can be envisioned that vary between these two settings and which may play a role: The grain size composition (see general comment above), oxygen abundance, and changes in, e.g., temperature and light conditions. A discussion about the implications of these differences would be a welcome inclusion.

R1AR9 – Justification for the removal of the coarse fraction has been made. McLagan et al. (2022) and Bollen et al. (2008), both show this is a consistently oxic aquifer. In terms of light, the experiments were performed in a basement lab and lights turned off whenever staff were not in attendance. There is minor potential for photochemical reductions on materials at the surface of the columns as the columns were transparent (opaque columns were considered but dismissed due to the inability to monitor the columns for breakup and/or preferential flow paths). Temperatures in the aquifer were consistent between 12 and 14 deg C, while temperatures in the lab were around 20 degC. Again, there could be some minor influence from the 6-8 degC on reductions.

To be clear, column experiments do provide a more representative reconstruction of real world conditions than batch experiments, which our group have already conducted. Nonetheless, they are still not the actual conditions, and we would like to highlight here that these differences between lab experiments (batch and columns) and real world conditions are very well known and heavily discussed within the literature over many decade (perhaps close to a century) of research on movements of analytes in soils and aquifer (and briefly in the introduction to the manuscript). Therefore, we do not feel another in-depth discussion on these limitations is warranted.

One final point here is that our research group is one of the only groups to focus on the study of mercury dynamics in aquifers in "the real world" (along with the group of Dr. Carl Lamborg). We have collected, analysed and published extensive field data from groundwater wells and soil cores from this and other sites. Yet these data themselves are not entirely conclusive (as is the case with all environmental field studies). Thus, it is critical that we use other tools such as column experiments to complement field studies and generate more holistic, total systems assays to improve our understanding of mercury biogeochemistry in soil-groundwater

systems, which is the ultimate goal of this work (and these field studies); an objective that the reviewer themselves in their opening paragraph have stated that we have achieved here.

R1C10 - Furthermore, the calculations for the back-of-the-envelope calculation could be included in the supplementary information to clarify how this estimation was done.

R1AR10 – Figure S9.6 and Table S9.1 will be added to the SI showing the details (including the percentage of the integrated area of the peaks; 0.11% Hg(0)) for Hg(0) and Hg(II) peaks from the mean PTD curves for all columns from EXP2. The text referring to the *back-of-the-envelope* calculations has been updated as follows:

"If we conservatively assume conservative values fora mean depth of contamination of 2 m (aquifer ≈3-4 m depth; Bollen et al., 2008; McLagan et al., 2022), mean THg concentration of 2 mg kg-1 (solid phase THg concentration of 2-162 mg kg along the contaminated aquifer; Bollen et al., 2008), the fraction of Hg(0) produced per day is 0.01 – 0.001% of the THg (based off 0.1% Hg(0) peak integration of total peak area of mean PTD curve from EXP2; see Section S9) based off data from Bollen et al., 2008; McLagan et al., 2022), and the same bulk density and flow rates as in our experiments, we can produce a back-of-the-envelope estimate of the mass of Hg(0) produced and potentially lost from the aquifer to overlying soils."

R1C11 - S1, Background Investigations section: It is stated that the equipment was tested for DOC to investigate the origin of the discoloration. What was the result of these tests?

R1AR11 - "DOC measurements from deionised water leaches (after initial rinsing all equipment with dionised water) were below limits of detection." This sentence will be added to Section S1.

*Technical Corrections*

R1C12 - Ln 116: Remove the "from".

R1AR12 – This will be changed to "source from an area adjacent to a former…"

R1C13 - Ln 155: "Names" should be "named."

R1AR13 – will be changed.

R1C14 - Ln 246: The protocol summary makes more sense if the relative rather than the absolute volume of BrCl in the modified aqua regia is specified.

R1AR14 – The part in parenthesis will be changed to "(8 mL HCl, 3 mL HNO3, and 1mL BrCl)".

R1C15 - Ln 338: Insert "to" in "species used to generate stock solution".

R1AR15 – will be changed.

R1C16 - Figure 2: The dashed lines are faint, and the color contrasts between the Freundlich and exponential decay functions are hard to discern. It is suggested that the figure be edited to make it easier to interpret.

R1AR16 – Figure 2 quality and clarity will be updated to reflect this request. Similarly, at the request of the associated editor, the quality/resolution of all figures has been improved.

R1C17 - Ln 364: "EXP" should be "EXP1".

R1AR17 – will be changed.

R1C18 - Ln 394: "recovers" should be "recovery".

R1AR18 – will be changed.

R1C19 - Ln 407: A period sign is missing.

R1AR19 – will be changed.

R1AR20 - Ln 428, Ln 476: The use of "that" or "this" in the first sentence of the paragraph, referring to the end of the previous one, impacts readability.

R1AR20 – updates to these sentences will be made accordingly.

R1C21 - Ln 434, 495: "Overtime" should be "over time".

R1AR21 – will be changed.

R1C22 - Ln 464-464, figure 3: It is suggested to label the panels in Figure 3 a) and b) and make the corresponding references in the text.

R1AR22 – Panel Labels will be added.

R1C23 - Ln 567: "...that fraction..." should be "...that the fraction...".

R1AR23 – will be changed.

R1C24 = References: The Miretzky 2005 reference is not listed.

R1AR24 – will be added.

R1C25 - S1: There are references to Appendices A 3 and A 4, but no appendix is included apart from the supplementary information. It appears that the S1:3 and S1:4 figures, respectively, are what is meant to be referenced.

R1AR25 – will be changed.

R1C26 - Fig S1.5: While not critical for a SI figure, the figure could be improved by shortening the X-axis (to 100 instead of 180 minutes).

R1AR26 – This will be updated to reflect the suggestion.

R1C27 - S1: The "results of the preliminary test" section: The takeaway, aided by figure S1.6, is that the concentration was ramped to reach a concentration where the eluate concentration was high enough for the experiment. This is, however, not clear when reading the text and should be clarified.

R1AR27 - The figure caption states: "Three different initial solutions were tested. In the first section (white), a solution with 1.7 ± 0.2 mg L-1 was used, in the second section (blue) 7.5 ± 0.8 mg L-1 and in the last section (violet) 49.3 ± 4.4 mg L-1."

The text states: "The pre-experiment was divided into three sections, where the initial solution had different concentrations (1.7 ± 0.2 mg L-1, 7.5 ± 0.8 mg L-1, and 49.3 ± 4.4 mg L-1). At this stock solution concentration, the increase in the liquid phase THg concentration was very little after 10 days. Hence it was deemed too low, and the stock solution increased. At 7.5 ± 0.8 mg L-1, the concentration in the liquid phase reached 32.8% of the maximum possible concentration of 7.5 ± 0.8 mg L-1 of the initial solution after four more days, despite 9 L of 1.7 ± 0.2 mg L-1 already being added to the columns. Thus, the stock solution concentration was increased again to 49.3 ± 4.4 mg L-1."

We are unsure that we can be much clearer on this, especially when the reviewer themselves are stating that they understand?

R1C28 - S2: Table S2.2 header "Elements" should be specified to clarify that wavelengths are listed below.

R1AR28 – will be changed.

R1C29 - S2: "Table S1.5" should be "Table S2.5".

R1AR29 – will be changed.

REFERENCES:

Lewis, J., & Sjöstrom, J. (2010). Optimizing the experimental design of soil columns in saturated and unsaturated transport experiments. *Journal of contaminant hydrology*, *115*(1-4), 1-13.

Pan, B. C., Meng, F. W., Chen, X. Q., Pan, B. J., Li, X. T., Zhang, W. M., ... & Sun, Y. (2005). Application of an effective method in predicting breakthrough curves of fixed-bed adsorption onto resin adsorbent. *Journal of Hazardous Materials*, *124*(1-3), 74-80.

Cucarella, V., & Renman, G. (2009). Phosphorus sorption capacity of filter materials used for on-site wastewater treatment determined in batch experiments–a comparative study. *Journal of environmental quality*, *38*(2), 381-392.

Zhu, T., Jenssen, P. D., Maehlum, T., & Krogstad, T. (1997). Phosphorus sorption and chemical characteristics of lightweight aggregates (LWA)-potential filter media in treatment wetlands. *Water Science and Technology*, *35*(5), 103-108.

Bollen, A., Wenke, A., & Biester, H. (2008). Mercury speciation analyses in HgCl2-contaminated soils and groundwater—implications for risk assessment and remediation strategies. *Water Research*, *42*(1-2), 91-100.

McLagan, D. S., Schwab, L., Wiederhold, J. G., Chen, L., Pietrucha, J., Kraemer, S. M., & Biester, H. (2022). Demystifying mercury geochemistry in contaminated soil–groundwater systems with complementary mercury stable isotope, concentration, and speciation analyses. *Environmental Science: Processes & Impacts*, *24*(9), 1406-1429.

---

## Author Comment (AC2)

*We use the following format for comments and responses:*

*R2C1 = Reviewer 2, comment 1. R2AR1 = Reviewer 2, Author Response 1.*

R2C1 - I first congratulate the lead scientist and the entire team. The manuscript offers interesting findings providing valuable information for Hg research, especially on Hg mobility. The manuscript is well-documented and also well-written, particularly in explaining the experimental methods. I recommend this manuscript to be published in SOIL with minor suggestions.

R2AR1 – These are very kind comments and we thank the reviewer for their time and support of our work.

R2C2 - Abstract: The conciseness of the abstract can be improved. Some details can be moved to the conclusion section, ensuring the main finding remains highlighted.

R2AR2 – Briefly, the reason a conclusion section was not included as it is quite often repetition of an abstract, which serves to summarise the manuscript, and statements and suggestions made in the results and discussion section. Thus, it is our preference to not include a conclusion section for conciseness and to prevent unnecessary repetition. If the associate editor insists on a conclusion section it can be added.

Nonetheless, the abstract will be improved for conciseness and to remove some redundancies in the resubmitted manuscript.

R2C3 - Lines 53-66: The first paragraph can be re-arranged, some of the sentences are too broad and have little link to the research.

R2AR3 - [This is the same comment as R1AR3 and comes from leading author] I thank the reviewer for their suggestion (and this is a similar suggestion to reviewer 1). However, I deem this to be more related to preference of writing style. I prefer to introduce my papers with something different that provides unique information that some readers may learn and benefit from rather than repeating the same information that is stated in most every paper on a particular topic. For instance, I would ask how many papers has the reviewer read on mercury that have included some form of "mercury is a persistent, bioaccumulative, and toxic contaminant…" (or some other very well noted anecdote on mercury) within the first few sentences? Here, I have introduced information on why mercury is such a unique metal (its quantum/electronic structure), information that is not well known even to many mercury scientists.

Moreover, I would argue that this has quite important "links to the research" as the behaviour of mercury in such saturated environments is influenced greatly by the quantum/electronic structure. For instance, the volatility of elemental mercury, which we link as a potential loss process of Hg from groundwater (and surface soils and water) is directly linked to elemental mercury not forming metal-metal bonds. It is my stylistic preference that the introduction remain in this form; I do feel strongly that individual writing styles still very much have a space and a place within scientific writing.

R2C4 - Lines 122-124: is there any information on geologic materials and the site description where the samples were taken?

R2AR4 – The following sentence will be added to the methods section:

"The geology and structure of the soil/aquifer profile has been described in detail in previous works (Schöndorf et al., 1999; Bollen et al., 2008; McLagan et al., 2022)."

R2C5 - Line 140: put a space, "8 x 60 mL ..."

R2AR5 - will be changed.

R2C6 - Line 143: DI = deionised?

R2AR5 - will be changed to "deionised".

R2C7 - Line 148: what is the desired volume or target density?

R2AR7 – We followed guidelines presented by Lewis & Sjöstrom (2010) for these aquifer materials which have been defined as sandy-gravel. The following sentence has been added to the methods:

"According to Lewis and Sjöstrom (2010), the average bulk densities range from 1.2 – 2.0 g cm3 for sands and 1.6 – 2.0 g cm3 for gravel. Thus, we deem the achieved bulk density of the columns to be appropriate for these materials, particularly as densities of the removed coarse materials are higher (solid densities are estimated at 2.65 g cm3; Lewis and Sjöstrom (2010))."

R2C8 - Lines 171-173: The difference between EXP1 and EXP2 can be placed at the beginning of the paragraph, so readers can directly point out the difference between the two experiments.

R2AR8 – This is a good suggestion. This long paragraph will be broken up and a new paragraph will be started at "Stock solutions…". As reviewer 2 suggests, the difference between the experiments will be better highlighted at the beginning of a paragraph in the updated manuscript.

R2C9 - Lines 172-173: The amount of HgCl concentration used in the experiment was estimated by considering the current concentration (after 55 years). Is there any data or information about the loss of Hg concentration (about 1000x) over 55 years?

R2AR9 – [this comment is the same as R1AR4] The original concentration of the solution applied during the kyanisation process was 0.66% or 6600 ppm HgCl2 solution. Contamination at the site was from spills of this solution, which contains the very soluble HgCl2. Thus, we know solution ~50x stronger the EXP2 and >100x stronger than EXP1 were entering the top of the soil profile and this site operated from 1904-1965 with 10-20 T of Hg being lost to the soils and aquifer. These data are stated in the study and other referenced work examining this site. Without question, experimental constraints (time) was a major factor in selecting these experimental stock solution THg concentrations, but considering the history of the site we do deem them to be applicable to this site and others. This is supported by the fact that Miretzky et al., 2005 (the only other study Hg column experiments) applied similar concentration to a similar column experiment design. This paragraph has been updated as follows to better describe these considerations:

"Stock solutions were 46.1 ± 0.1 mg L-1 in EXP1 (n = 6) and 144 ± 6 mg L-1 in EXP2 (n = 12) and were selected for (i) time considerations (see Figure S1.6) and (ii) these values remain between HgCl2 concentration applied during industrial activities (6600 mg/L; spillages of this solution to the top of the soil profile) and recently measured groundwater concentrations up to 164 ± 75.4 μg L-1 observed 55 years after cessation of the industrial activities at the site (McLagan et al., 2022)."

R2C10 - Figure 2: The blue and red dashed lines are not clearly seen.

R2AR10 – Figure 2 quality and clarity will be updated to reflect this request. Similarly, at the request of the associated editor, the quality/resolution of all figures has been improved.

R2C11 - Line 364: should be EXP1

R2AR11 - will be changed.

R2C12 - Lines 369: Reference of Miretzky et al. (2005) is missing

R2AR12 - will be added.

R2C13 - Lines 377-379: The authors pointed out the potential role of clay minerals or Fe/Mn oxides as an important solid-phase for the sorption of soluble Hg. In this case, the authors provide the properties of the solid-phase aquifer, such as Fe, Mn, clay, silt, sand (Table 1), and metal cations (S2). The pH was neutral to slightly basic, which support the adsorption of Hg into inorganic material. However, the clay content is low (table 1). Perhaps the authors could also provide a direct

link/evidence between the Hg sorption and the specific minerals or metals involved in this process.

R2AR13 – The clay content is 13.5%, which, along with Fe and Mn oxides present in sand/silt materials would provide sufficient mineral surfaces for sorption of all Hg applied within the columns. We do not have details on specific mineralisation of these materials and adding such here would be purely speculative. We do not believe such speculation would benefit the manuscript.

R2C14 - Conclusion: I suggest the authors include a conclusion section. The conclusion can contain a more comprehensive summary and suggestions for future work.

R2AR14 – Please see R2AR2.

REFERENCES:

Lewis, J., & Sjöstrom, J. (2010). Optimizing the experimental design of soil columns in saturated and unsaturated transport experiments. *Journal of contaminant hydrology*, *115*(1-4), 1-13.

---

## Author Comment (AC3)

*We use the following format for comments and responses:*

*R3C1 = Reviewer 3, comment 1. R3AR1 = Reviewer 3, Author Response 1.*

R3C1 - The manuscript "Organic matters, but inorganic matters too: column examination of elevated mercury sorption on low organic matter aquifer material using concentrations and stable isotope ratios" uses careful mercury breakthrough column experiments to determine the transport of mercury in low organic matter aquifer material. The experiments and analytical work are well done, and I commend the authors on their careful work. The information contained within the manuscript, particularly the role of inorganic materials in mercury transport, will be of interest to the wider mercury scientific community.

R3AR1 – We thank the reviewer for their time and overall positive sentiments towards the manuscript.

R3C2 - However, the lack details of the solute transport modelling methods raises more questions and concerns that need to be addressed, as detailed below.

R3AR2 – We will attempt to address and allay these concerns below and within the resubmitted manuscript.

**Specific Comments**

R3C3 - The methods detail the experimental methods well but do not detail the fitting of the solute transport model to the data. The lack of such section (and associated results) causes confusion as the breakthrough curves presented in figure 2 are dependent on time, while the equation presented in Table S6.1 is not dependent on time. How did the authors fit the equation to the data? Was the advection-dispersion equation fit first using the chloride tracers to estimate conservative transport parameters (i.e., hydrodynamic dispersion of the media) then the Freundlich equation fit using the Hg data? Or were the presented equations in S6 fit to the data without considering flow (i.e., a statistical fit). If the latter, then the resulting parameters have no physical meaning and are only the summation of multiple processes occurring (advective transport, surface complexation, mineral (clay) matrix diffusion, reduction to Hg0, desorption, etc.), thus not representing the Freundlich isotherm as suggested in the text (this point also applies to the flushing analysis and the exponential decay function). I also note that the Freundlich equations presented in S6 deviates from the common form ($q_e=K_F*C_e^{(1/n)}$), where qe is the solid phase concentration, Kf is the Freundlich equilibrium constant, Ce is the aqueous concentration and n is a fitting constant determined by linearizing the equation. There was no citation for the specific Freundlich equation used in the text, nor for the reasoning behind dividing the breakthrough curve into two separate analyses. The resulting units of the proposed curves are mg/L$^2$ but the Freundlich isotherm units (Kf) are mg/g (mass/mass).

Fitting the advection-dispersion equation with a Freundlich or Langmuir (as discussed below) isotherm is straightforward in freely available programs such as RETC or Hydrus-1D. There needs to be a clear section on the modelling procedure, including assumptions and equations, since the results and discussion rely on these methods. This needs to be added to the main manuscript prior to publication.

R3AR3 – Firstly, we want to clarify that the curves presented are breakthrough curves and not sorption isotherms. So reference to sorption isotherms functions does not apply here. A sorption isotherm would plot the solid phase concentration against the concentration within the eluate. We cannot plot sorption isotherms using our data as we only have ~50% breakthrough and equilibrium solid-phase concentrations. We did not have sufficient mass of solid phase materials to conduct enough experiments to sacrifice sufficient columns during breakthrough to create sorption isotherms (we note again here that we used all available solid phase materials in these experiments; none of this material remains). We could create sorption isotherms theoretically based on what we would expect in the solid phase from the eluate concentrations (i.e., using data from Table 2). However, this would essentially be plotting the liquid-phase concentration data against itself (the "theoretical" solid-phase concentrations are derived directly from the liquid-phase concentration). This approach would add nothing to the study; and hence, we do not consider it valid to derive sorption isotherms for these experiments.

Nevertheless, the Freundlich and exponential decay functions (/models) were fitted empirically. The purpose of this was to create a fit that displayed the behaviour of the data (not attempting to define mechanism from the function). Considering the comments from reviewer 1 and 3, we are prepared to take two course of action: (1) simply remove both sets of functions and let the data itself display the "S-shaped" breakthrough profile and "exponential decay-like" desorption profile; or (2) leave the fitted functions in, and more clearly define that these are empirically (and not mechanistically) fitted functions that serve to assist the readers in understanding the data. To be very clear, we derive the mechanisms of sorption and desorption from the concentration data and the stable isotope data, not the fitted functions. We are happy for the associate editor to specify which method they deem most appropriate.

Our preferred option is option (2) (leaving the fitted functions in). With this option, we will update the first sentence of Section 3.1.1 and the caption of Figure 2 to note the functions are empirically fitted more clearly:

"As expected, the uptake of the HgCl2 solution to the solid phase aquifer materials followed an S-shaped breakthrough curve best described by a Freundlich function (Figure 2; note these are empirically fitted functions)."

"…Sorption curves were fitted with Freundlich functions (blue dashed lines), and desorption curves were fitted with exponential decay functions (red dashed lines).

These functions were empirically (not mechanistically) fitted to the data as these plots are not sorption isotherms (see Section S6 for details of fitting functions)."

If the editor prefers option (1) (to remove the functions) we will remove the functions and references to the specific named functions. The reason that this is our less preferred option is that this is purely for naming purposes; this has no impact on our description of the sorption and desorption mechanisms (based on concentration and stable isotope data themselves). Using these empirically (well)-fitted functions ($R^2 > 0.99$ in all cases) makes descriptions more efficient and the story more concise.

**Other Specific Comments**

R3C4 - L27-28 The Freundlich model describes sorption not breakthrough curves, which is mediated by water flow. This needs to be more clearly stated here, perhaps adding "of the" in-between sorption and breakthrough.

R3AR4 – Yes, we agree. As we described in R3AR3, the Freundlich function was empirically fitted empirically and not for mechanism (not a sorption isotherm).

R3C5 - L61 comma splice after solubility. Regardless of this small grammatical issue, perhaps changing "have" to "having" will help the readability of the sentence.

R3AR5 – This will be changed to "…but are relatively stable ores that have very low solubility and bioavailability…"

R3C6 - L201-202 What is the total volume taken for analysis (was it the same as the waste 10mL or was the 15mL tube mentioned in 2.2.3 filled) and how does this compare to the total pore volume? If the ratio between the sample volume and the pore volume is large, then there needs to be some consideration that a sample taken at a given time represents a range of times rather than a specific time as assumed in the analysis. The larger the ratio the less precise your results. This comment also applies to the Hg isotope samples.

R3AR6 – 5 mL of sample was collected, which represents <25% of one pore volume. The following sentences was updated in Section 2.1 to reflect this:

"10 mL of eluate was allowed to flow off into a waste vessel before 5 mL of sample was collected periods for analysis (this applied to all analyses)."

We also note that we consider the different volumes sampled between columns as the x-axis uncertainty in the boxplot data (Figure 2).

R3C7 - L296-297 The definition of effective porosity is the proportion of total void space that is capable of transmission fluid under advective fluxes. In most cases,

this value is close to the total porosity but some media it can be quite a bit lower. Given your packed columns, I suspect that your effective porosity is closer to your total porosity as you defined. However, this assumption needs to be explicitly stated or measurements of effective porosity (e.g., soil air content at -100 mb) presented to confirm your assumptions.

R3AR7 – We added the sentence "effective porosity (…; assumed to be equal to total porosity)"

R3C8 - L301-302 Given the well-known soil texture and artificially packed nature of the columns, the effective porosity can be relatively accurately estimated using freely-available pedotransfer functions (Rosetta — ISMC (soil-modeling.org)). Such estimations would allow for KD to be estimated on all columns that achieved 50% breakthrough. These values can then be compared to your estimated values from columns C1.1-1.3 and C2.1-2.3.

R3AR8 – One of the major goals of this study was to simulate the original contamination to understand mechanisms of Hg dynamics in low OM aquifer materials and then derive Kd and Rd values from these column experiments for future modelling work (these experimentally derived Kd and Rd values are completely lacking). We deem the additional modelling scenarios suggested here by reviewer 3 outside the scope of this work and a part of the aforementioned future modelling work that applies the data obtained from these experiments.

R3C9 - L311 missing closing parenthesis.

R3AR9 – This will be fixed in the updated manuscript.

R3C10 - L303-316 How much Hg could be sorbed to the walls of the syringe? Would accounting for this improve your Hg mass balance (increasing eluate concentration from ≈91% for instance)? I note you discuss this briefly, but this may be worth a small batch experiment to see if the amount of sorbed Hg onto the plastic might be significant.

R3AR10 – As we have stated, there is no more of this aquifer material remaining. We had a limited supply of this material (it was donated to us from our colleagues who made the drill core), and it has been consumed. Hence, additional batch experiments with these materials is not possible. Simply doing this "batch" scenario without any materials (zero flow) is an entirely different scenario to a column of packed solid-phase materials with materials filling "roughened" surfaces of the syringes and impacting/limiting flow. We do not consider that such experiments would reveal any additional information that would benefit the study.

R3C11 - L364-365 The determination of theoretical max sorption condition suggest that the sorption characteristics can be fit with a Langmuir isotherm rather than a

Freundlich isotherm model. Based on the discussion and proposed multi-mechanism for Hg desorption (outer-sphere complexation vs mineral matrix diffusion), a multi-site Langmuir adsorption isotherm may best describe the actual processes rather than the simpler Freundlich isotherm. The authors seem to use a 2-site exponential decay function in the Desorption phase of EXP2 but no discussion of this is presented nor the rational. By using the Freundlich isotherm it is assumed that both processes that remove Hg from the liquid phase (complexation and mineral matrix diffusion) are occurring at the same rate and have the same potentials. The flushing phase of your column experiment and the isotopic results, with the current analysis, suggest that this is not the case. See Swenson and Stadie (2019, 10.1021/acs.langmuir.9b00154) for a good overview of Langmuir isotherms.

R3AR11 – We refer the reviewer to the explanation given in R3AR3.

R3C12 - L574 I appreciate the back-of-the-envelope calculations that really contextualize the magnitude of the potential Hg0 production at the contaminated site, however there needs to be a more explicit description of the mathematics and the values used in the calculation either here or in the SI.

R3AR12 (Please not this is the same response provide in R1AR10) – Figure S9.6 and Table S9.1 will be added to the SI showing the details (including the percentage of the integrated area of the peaks; 0.11% Hg(0)) for Hg(0) and Hg(II) peaks from the mean PTD curves for all columns from EXP2. The text referring to the *back-of-the-envelope* calculations has been updated as follows:

"If we conservatively assume conservative values fora mean depth of contamination of 2 m (aquifer ≈3-4 m depth; Bollen et al., 2008; McLagan et al., 2022), mean THg concentration of 2 mg kg-1 (solid phase THg concentration of 2-162 mg kg along the contaminated aquifer; Bollen et al., 2008), the fraction of Hg(0) produced per day is 0.01 – 0.001% of the THg (based off 0.1% Hg(0) peak integration of total peak area of mean PTD curve from EXP2; see Section S9) based off data from Bollen et al., 2008; McLagan et al., 2022), and the same bulk density and flow rates as in our experiments, we can produce a back-of-the-envelope estimate of the mass of Hg(0) produced and potentially lost from the aquifer to overlying soils."

R3C13 - L592 Given the reactive transport focus of the paper, I suggest replacing "three-dimensional spread" to the more appropriate terms "longitudinal and transverse dispersion".

R3AR13 – This will be updated in the next version of the manuscript.

R3C14 - L598 I really like this point!

R3AR14 – We thank the reviewer for their kind words. We do believe this also links directly to the next comment (R3C15).

R3C15 - L600-615 The lower, or even different, KD for the higher concentration EXP2 is somewhat surprising, as the KD is the linear partitioning coefficient, which assumes equilibrium between the liquid and solid phases. Assuming there is no saturation of adsorption sites, KD should be close if not equal in both experiments since the materials, packing, and flow rates are the same. However, there was no explanation of these values or their differences beyond stating other literature values. Are the differences in KD due to slight variations in clay content (experimental error) or is there another explanation? I suggest that there needs to be a bit more discussion here to explain these values more mechanistically.

R3AR15 – We did provide an explanation for the difference in Kd values between EXP1 and EXP2:

"The difference in RD and KD values between EXP1 and EXP2 (Table 3) indicate stock solution concentration is a factor in the transport of mercury within these columns. The elevated stock solution concentrations may be undermining the assumption of equal accessibility to sorption sites; REDUCED KINETICS AT HIGHER CONCENTRATIONS (USEPA, 2004)."

The part in ALL CAPS will be added for clarification purposes within these sentences.

However, as we note it was our intention to simulate the original contamination, and we would also refer to the reviewers support of our statement (R3C13) in L598:

"Considering the high concentrations of Hg that have been observed within this and other Hg contaminated aquifers (Katsenovich et al., 2010; Lamborg et al., 2013; Demers et al., 2018), it is critical that we do not isolate our study of Hg transport dynamics to low concentration experiments that meet assumptions for theoretical sorption (batch and column) experiments."

---

## Referee Report (RR1)

Reviewer report:

I commend the authors on revising the manuscript and answering my previous comments pleasingly. I have no further questions or concerns, and I would recommend the manuscript be published in SOIL.

When reading through the revised manuscript, I only found one typographic error:

Ln 593: reinsert the 'a' in "Conservative values for mean depth of contamination".